

**Classifying offshore faults for hazard assessment: A new**
**approach based on fault size and vertical displacement**
May Laor[1,2*], Zohar Gvirtzman[1,2]
[1] Geological Survey of Israel, Yesha'yahu Leibowitz 32, Jerusalem, Israel
[2] Institute of Earth Sciences, The Hebrew University of Jerusalem, Israel
* Correspondence to: May Laor (may.laor@mail.huji.ac.il)
**Highlights**
• Mapping "active faults" offshore for hazard assessment is a challenge that
frequently ends without an answer.
• Utilizing high quality seismic data, we suggest a new approach for master
planning.
• Based on the recent displacement and fault plane size, we classify faults to 3
hazard levels.
• Large faults scarps in an area of fast sedimentation indicate seismic rupture
rather than creep.



## Abstract

For many countries, the methodology for offshore geohazards mitigation lags far
behind the well-established onshore methodology. Particularly complicated is the
mapping of active faults. One possibility is to follow the onshore practice, i.e.,
identifying a sub-seabed Holocene horizon and determining whether it displaces this
horizon for each fault. In practice, such an analysis requires numerous coring and often
ends without an answer.
Here we suggest a new approach aimed for master planning. Based on high-quality
seismic data, we measure for each fault the amount of its recent (in our specific case
350 ky) displacement and the size of its plane. According to these two independently
measured quantities, we classify the faults into three hazard levels, highlighting the
"green" and "red" zone for planning.
Our case study is the Israeli continental slope, where numerous salt-related, thin-
skinned, normal faults dissect the seabed, forming tens of meters high scarp, which are
crossed by gas pipelines. A particular red zone is the upper slope south of the Dor
disturbance, where a series of big listric faults rupture the seabed in an area where the
sedimentation rate is four times faster than the displacement rate. We suggest that this
indicates seismic rupture rather than creep.



# 1. Introduction

## *1.1. Marine geohazards*

The need for geohazards assessment in the marine environment is increasing globally due to the growing number of infrastructures laid on the seafloor. To mitigate marine geohazards, numerous studies have been conducted in many world basins (Georgia Basin, (Barrie et al., 2005); Sea of Marmara (Armijo et al., 2005); Gulf of Mexico (Prior and Hooper, 1999; Angell et al., 2003); offshore California (Clark et al., 1985, and the ref in); Norwegian Sea (Shmatkova et al., 2015); Italian Sea (Chiocci and Ridente, 2011), and more). Most of these studies focus on submarine landslides and when faults are considered, they are commonly treated as static seabed obstacles. Note, however, that even extremely accurate mapping of the seafloor does not provide information about the possibility of rupture or creep. For this, displacement of dated horizons at the subsurface should be measured utilizing high-resolution seismic surveys and core analyses (Posamentier, 2000; Kvalstad, 2007; Hough et al., 2011). In general, site investigation for faults hazard offshore includes four steps (Prior and Hooper, 1999; Angell et al., 2003): (a) Mapping the seafloor, (b) establishing a chrono-stratigraphic scheme by tying high resolution seismic data to dated horizons in boreholes, (c) structural mapping of the fault and displacement measurements, (d) geological interpretation and quantification. The difficulty in site surveys is that each of them requires months of work and frequently yields uncertain results. In many cases the displaced horizons are too deep to core (high sedimentation rates) or too shallow to be detected seismically (low sedimentation rates). In practice, the preparation of active fault maps (as well as other hazard maps) for offshore areas is lagging decades behind the onshore environment.



### 1.2. Goal

The goal of this study is to provide a practical and relatively fast solution for early-stage planning of marine infrastructure. Instead of answering a yes-and-no question (active or not) for each fault traced on the seabed, we classify all faults to three hazard levels highlighting "green" and "red" zones for master planning. Instead of searching for displaced horizons that are younger than 11 ky (Bryant and Hart, 2007) by trenching or coring, we take advantage of the high quality seismic data frequently collected offshore. Instead of investing huge efforts (multiple coring to a dated horizon) in finding whether or not each specific fault in the study area meets a pre-defined criterion of 'activeness', we map the subsurface and determine the levels of fault activity based on the amount of recent displacement and the size of the fault plane.

The case study analyzed here is the Israeli offshore (Fig. 1), where numerous faults dissect the continental slope. Theses faults are related to thin-skinned salt tectonics and are associated with tens of meters high seabed scarps. To measure the recent vertical displacement, we start with measuring heights of seabed fault scarps and continue with measuring vertical displacements of a subsurface horizon dated by Elfassi et al. (2019) to 350 ky. To measure fault size, we map fault planes in the subsurface or, at least, fault length in map view. When 3D mapping is possible, we distinguish between small stand-alone surface faults and small surface segments that connect at the subsurface and form much bigger faults planes.

In addition, we distinguish between three fault groups that differ in their location (i.e., proximity to salt wedge and continental slope) and structure (i.e. steepness), allowing further evaluation of the results in light of faults mechanisms.



## 2. Scientific background

### 2.1. Geological history of the Levant Basin

The Levant Basin was formed in the late Paleozoic and early Mesozoic, alongside the opening of the Tethys Ocean that had separated Africa plate from Eurasia plate (Garfunkel, 1984, 1988, 1998; Robertson, 1998). At that time, several rifting phases created a system of horsts and grabens spreading from the northern Negev north-westwards into the Levant basin (Bein and Gvirtzman, 1977; Garfunkel, 1984, 1988, 1998; Robertson, 1998). After the rifting stage, approximately at the end of the Early Jurassic (~180 Ma), the Levant continental margins turned passive and continued to accumulate sediments for more than 100 million years (Gvirtzman and Garfunkel, 1997, 1998; Steinberg et al., 2008; Bar et al., 2013).

At the end of the Turonian and the beginning of the Santonian (~84 Ma), a change in the relative movement between Africa and Eurasia led to a change in the stress regime and folding along the "Syrian arc" began (Krenkel, 1924; Henson, 1951; De-Sitter, 1962; Freund, 1975; Reches and Hoexter, 1981; Eyal and Reches, 1983; Sagy et al., 2018).

About 35 million years ago, a large area including east Africa and northern Arabia, started rising above sea level. This process provided large amounts of clastic sediments to the Levant Basin, where the sedimentation rate increased significantly (Gvirtzman et al., 2008; Steinberg et al., 2011; Avni et al., 2012; Bar et al., 2013, 2016). These clastic sediments compose the Saqiye Group, which thickens from tens-hundreds of meters in the Israeli coasts to 1.5 km in the continental shelf area (Gvirtzman and Buchbinder, 1978), and 6 km in the deep Levant Basin (Steinberg et al., 2011).

About 6 million years ago, the connection between the Mediterranean Sea and the Atlantic Ocean was restricted during a short event termed the Messinian Salinity Crisis


(MSC). During the crisis sea-level dropped, and a few km thick evaporite sequence
accumulated in the entire Mediterranean Sea (e.g., Hsü et al., 1973; Ryan and Hsü,
1973). The salt sequence offshore Israel is nearly 2-km-thick in the deepest portion of
the basin, thinning landwards and nearly pinching out to zero beneath the continental
slope (Ryan and Cita, 1978; Mart and Gai, 1982; Gradmann et al., 2005; Bertoni and
Cartwright, 2006b; Netzeband et al., 2006b; Gvirtzman et al., 2013, 2017).
In the Pliocene, the Nile, one of the largest rivers in the world, supplied a huge amount
of sediments to the eastern Mediterranean that buried the Messinian salt and produced
a giant delta with a well-developed deep-sea fan (Mascle et al., 2001). Alongshore
currents transporting sediments from the Nile Delta through the Israeli coast gradually
formed the continental shelf offshore Israel (Gvirtzman and Buchbinder, 1978;
Goldsmith and Golik, 1980; Carmel et al., 1985; Stanley, 1989; Tibor et al., 1992b;
Buchbinder et al., 1993; Golik, 1993, 2002; Buchbinder and Zilberman, 1997; Perlin
and Kit, 1999; Ben-Gai et al., 2005; Zviely et al., 2006, 2007; Klein et al., 2007;
Schattner et al., 2015; Schattner and Lazar, 2016; Zucker et al., 2021). The slope of this
continental shelf is currently faulted by faults, which are the target of this study.

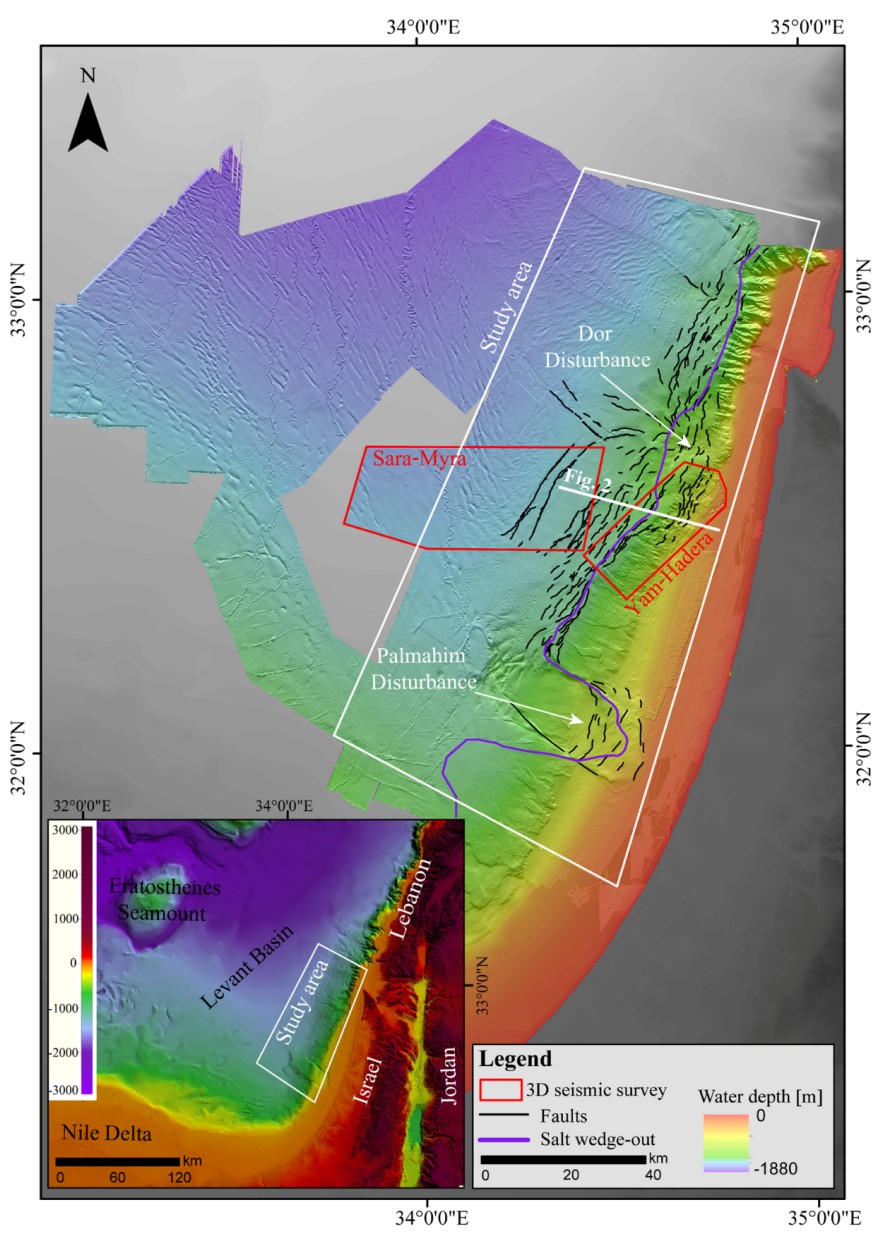

Figure 1: Location map. Bathymetry and faults (black lines) after Gvirtzman et al. (2015).
Ragional map from Hall (1994).


### 2.2. Thin-skinned salt-related normal faulting along the Israeli continental slope

Numerous thin-skinned normal faults rupture the seabed along the Israeli continental slope (Fig. 1), creating steep steps that are tens of meters high (Almagor and Garfunkel, 1979; Garfunkel et al., 1979; Mart and Gai, 1982; Almagor, 1984; Garfunkel, 1984; Garfunkel and Almagor, 1984; Tibor et al., 1992a; Gradmann et al., 2005a; Martinez et al., 2005; Bertoni and Cartwright, 2005, 2006b; Netzeband et al., 2006b; Mart and Ryan, 2007; Cartwright and Jackson, 2008; Cartwright et al., 2012; Gvirtzman et al., 2013b, 2015; Katz et al., 2015; Safadi et al., 2017; Kartveit et al., 2018; Gadol et al., 2019). Noteworthy, these fault scarps are not buried by sediments, indicating displacement rates higher than burial rates. On the other hand, averaged on hundreds of thousands of years, displacement rates are roughly similar to sedimentation rates (Elfassi et al., 2019a). This indicates that the fault scarps observed on the present seafloor may have formed by recent instantaneous seismic ruptures rather than by continuous creep (Elfassi et al., 2019a). Despite this seismic slip hypothesis, these relatively shallow thin-skinned faults are incapable of producing large earthquakes because their fault planes are relatively small compared to crustal faults. The major hazard they pose is surface rupture, which may as well trigger slumps (Katz et al., 2015).

The recognition that the faults along the Levant continental margin are related to thin-skinned salt tectonics has been stated in many studies (Neev et al., 1976; Ben-Avraham, 1978; Almagor and Hall, 1979; Garfunkel et al., 1979; Mart and Gai, 1982; Garfunkel, 1984; Garfunkel and Almagor, 1984; Tibor et al., 1992; Gradmann et al., 2005; Martinez et al., 2005; Bertoni and Cartwright, 2006a, 2007; Loncke et al., 2006; Netzeband et al., 2006a; Hübscher and Netzeband, 2007; Mart and Ryan, 2007; Hubscher et al., 2008; Cartwright and Jackson, 2008; Clark and Cartwright, 2009;





Cartwright et al., 2012; Gvirtzman et al., 2013; Gadol et al., 2019; Ben Zeev and
Gvirtzman, 2020; Hamdani et al., 2021). In particular, it has been suggested that
faulting was initiated by basinwards salt flow (Gradmann et al., 2005; Bertoni and
Cartwright, 2006b, 2015; Allen et al., 2016; Cartwright et al., 2018; Kirkham et al.,
2019) triggered by basinward tilting of the continental margin (Cartwright and Jackson,
2008; Elfassi et al., 2019; Hamdani et al., 2021).
The beginning of faulting was initially dated to a relatively wide time interval between
the late Pliocene and the early Pleistocene (e.g., Garfunkel et al., 1979; Almagor, 1984;
Garfunkel, 1984; Gradmann et al., 2005; Netzeband et al., 2006). Later, based on 3D
high-resolution seismic surveys, Cartwright and Jackson (2008) showed that offshore
central Israel faulting began in the mid-Pliocene; and then, in the late Pliocene it had
spread northward, and in the early Pleistocene southward.
Elfassi et al. (2019) established a new chronostratigraphic scheme for the Pliocene-
Quaternary section offshore Israel that allows better dating of faults. By combining
seismic and bio-stratigraphic data, they divided the Plio-Quaternary sequence into four
units (Fig. 2) : Unit 1- Pliocene (5.33-2.6 Ma); Unit 2- Gelasian (2.6-1.8 Ma); Unit 3-
Calabrian-Ionian (1.8-0.35 Ma); and Unit 4- Ionian-Holocene (<0.35 Ma). Based on the
improved Chrono-stratigraphy, Elfassi et al. (2019) measured displacement rates on
several faults offshore central Israel (in the Sara-Myra survey, Fig. 1), and concluded
that during the Pliocene faulting activity was minor (< 4 m/Ma), then, in the Gelasian,
it peaked to rates of >100 m/Ma (10 cm/ky), and later it decreased to rates of ~50 m/My
(5 cm/ky).




In what follows, we use the chrono-stratigraphy of Elfassi et al. (2019) to map the most
recent horizon (Ionian-Holocene, <350 ka) in the entire study area (light blue- base Unit
4 in Fig. 2b), and identify the zones with the strongest recent activity.

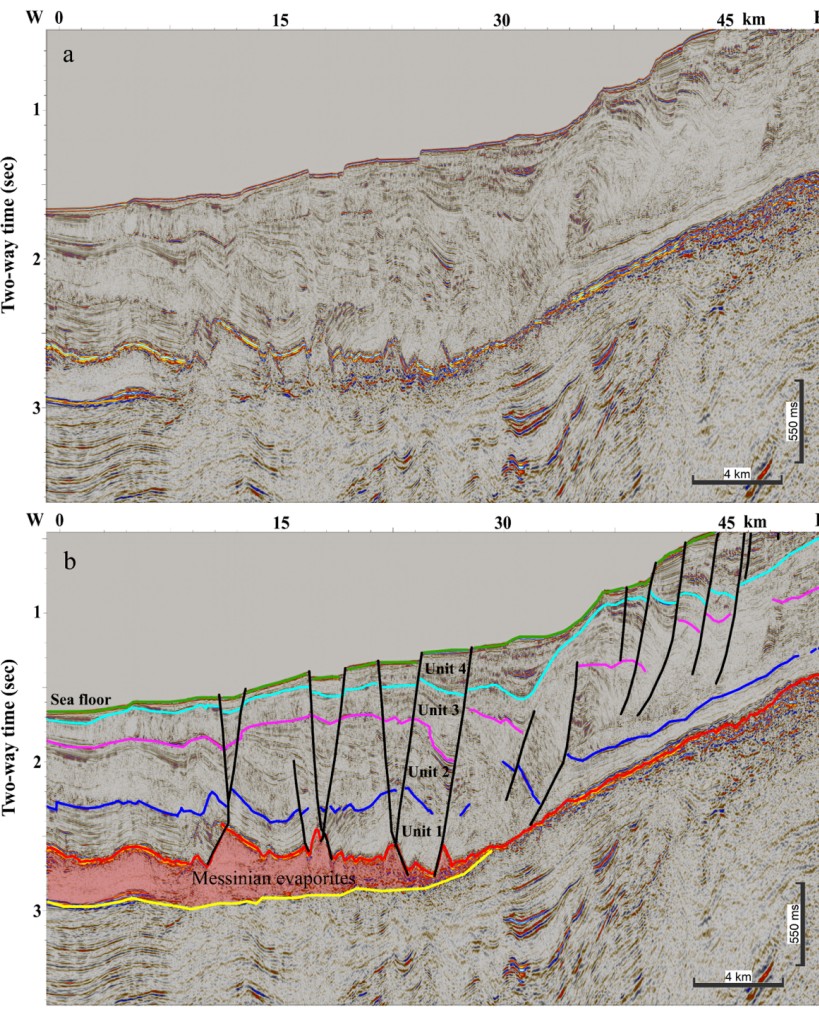


*Figure2 : Uninterpreted (a) and interpreted (b) seismic section across the Levant continental*
*margin offshore Israel (location in Fig. 1). Chrono- and seismo-stratigraphic of the Pliocene-*
*Quaternary section after Elfassi et al. (2019). Green- Sea floor, Light blue – base Unit 4, purple*
*– base Unit 3, blue – base Unit 2, red – base Unit 1 (and top evaporites), yellow – Base*
*evaporates. Thin-skinned faults in black lines.*


## 3. Data and Methods

### 3.1. Bathymetry

The Israel national bathymetry survey provides pixel resolution of 15 m until water depth of ~700 m (Sade et al., 2006, 2007) and 50 m between isobaths 700 m and 1700 m (Tibor et al., 2013). In addition, we used bathymetric grids with ~10 m cell size, derived from four 3D seismic surveys listed in Table 1 (Aviya; Dalit; Yam Hadera; and Sara-Myra).

To quantify the height of fault scarps at the present seafloor, we developed an algorithm that uses the fault map prepared by Gvirtzman et al. (2015) and automatically calculates elevation differences from both sides of the faults every 50 meters. This algorithm was applied to all grids described in Table 1.

### 3.2. Seismic interpretation

The seismic data used here include 2D and 3D seismic reflection surveys processed in the time domain (TWT), and 3D seismic cubes that were pre-stack depth migrated (Table 1). All surveys were loaded and interpreted using the Kingdom HIS software. Preliminary mapping of the four units described above was done by Elfassi et al. (2019). Ben-Zeev and Gvirtzman (2020) expanded this mapping to cover Israel's Exclusive Economic Zone (EEZ). Here we recheck and remap these horizons in detail along the continental slope where faults are common.

Subsurface mapping of faults adds several layers of information on top of seabed mapping: (1) it allows measuring the displacement of dated horizons, and thus indicates the rate of motion; (2) it allows distinguishing between small surface faults that are minor and small surface faults that connect at the subsurface to large faults; (3) it allows identifying hidden faults, which do not appear on the bathymetry, but may rupture it in





the future; (4) it provides a 3D view of the fault plane which is essential for structural

analysis and estimation of potential earthquake magnitudes.

*Table1 : Seismic data*

| # | Survey name | Survey type and units | Source | Survey's technical details | Grid cell size | Data available for this study |
|---|---|---|---|---|---|---|
| 1 | Aviya | Seismic reflection: Depth m | Delek Ltd. | Line spacing: 25 m x 12.5 m | 10 m | Bathymetry |
| 2 | Dalit | Seismic reflection: Depth m | Delek Ltd. | Line spacing: 25 m x 12.5 m | 10 m | Bathymetry |
| 3 | Yam Hadera | Seismic reflection: Depth m | Modiin Energy | Line spacing: 25 m x 12.5 m | 9 m | Seismic (3D), Bathymetry |
| 4 | Gabriela | Seismic reflection: Depth m | Modiin Energy | Line spacing: 25 m x 12.5 m | 13 m | Seismic (3D) |
| 5 | Sara-Myra | Seismic reflection: Depth m | Modiin Energy + ILDC | Line spacing: 25 m x 12.5 m | 10 m | Seismic (3D), Bathymetry |
| 6 | The Israel national bathymetry survey | Multibeam sonar: Depth m | (Sade et al., 2006; Tibor et al., 2013) | 15 m x 15 m till water depth of 700 m and 50 m x 50 m till water depth of over 1700 m. | 50 m, 15 m | Bathymetry |
| 7 | Isramco North Central | Seismic reflection: TWT sec | Isramco | Line spacing: 12.5 m x 12.5 m | | Seismic (3D) |
| 8 | TGS | Seismic reflection: TWT sec | TGS-NOPEC Geophysical Company | Shot interval: 25m Group interval: 12.5 m | 5-10 km | Seismic (2D) |




| | | | Total line length of ~6000 km. | | |
|---|---|---|---|---|---|
| 9 | HORIZON | Seismic reflection: TWT sec | Horizon Exploration Limited | Shot interval: 25 m | | Seismic (2D) |
| 10 | SPETRUM | Seismic reflection: TWT sec | Spectrum Energy & info. Tech. Ltd | Shot interval: 50 m<br>Group interval: 12.5 m<br>Streamer length: 7200 m | | Seismic (2D) |

## 4. Results

### 4.1.   Fault vertical displacement, sedimentation, and seabed scarps

Fig. 3a shows heights of seabed scarps measured from both sides of all faults every 50 m. The map shows that between the Palmahim and the Dor disturbances, fault scarps are relatively low (<20 m), whereas from the Dor disturbance northwards they are significantly higher (20-90 m). This observation is consistent with extension measurements that also increases northwards (Cartwright and Jackson, 2008; Ben-Zeev and Gvirtzman, 2020).

The problem with analyzing bathymetry alone is that faults scarps are reduced by sedimentation and erosion and do not correctly represent fault vertical displacement. Therefore, we also measure fault throw along the youngest regionally mappable horizon (base Unit 4, Fig. 3b), which yield displacement rates averaged for the past 350 ky (the best possible representation of 'recent' in the study area

This measure for recent vertical displacements highlights the vicinity of the Dor disturbance with the highest displacement rates reaching 40-50 cm/ky (Fig. 3b). This exceptionally active zone is not detected in the bathymetric analysis (Fig. 3a) emphasizing the need for subsurface measurements. To further illustrate the Dor anomaly, Fig. 3c shows a projection of all seabed and subsurface offset measurements



along a south-north section emphasizing peak throws near the Dor disturbance
(X~$3.6 * 10^6\ m$), nearly two times larger than in surrounding areas.

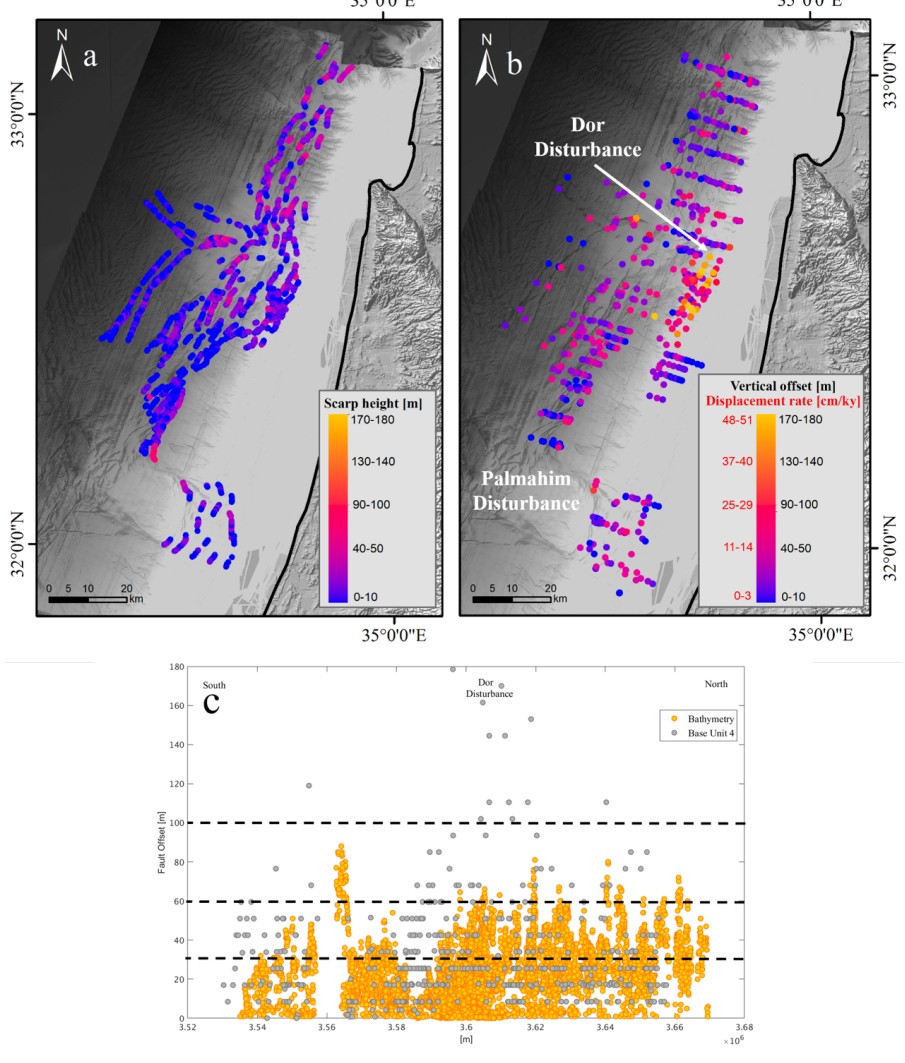


*Figure3 : Vertical offset measurements along faults. (a) Height of seabed scarps derived from*
*bathymetry analysis. (b) Vertical offsets at the base unit 4 horizon measured from seismic data.*
*Assigning 350 ky to the base Unit 4 horizon (Elfassi et al., 2019), its vertical offset is*
*transformed to displacement rate (the left hand side of the scale bar in b). (c) Vertical offset*
*measured at the base of unit 4 (gray dots) and scarps height at the seafloor (orange dots). Note*
*that vertical offsets in bathymetry increase northwards whereas vertical offsets at the base of*
*unit 4 increases in the vicinity of the Dor disturbance. Bathymetry from Tibor et al. (2013).*


Considering the 350 ky age of base Unit 4 (Elfassi et al., 2019), sedimentation rates
(thickness of Unit 4 divided by 350 ky) can be calculated for the entire study area (Fig.
4c). Results indicate relatively low (<60 cm/ky) values in the deep basin, increasing to
~90 cm/ky in the mid-slope and >150 cm/ky along the basinward propagating shelf
edge (Ben-Zeev and Gvirtzman, 2020). Particularly interesting is the off-shelf peak near
the Dor disturbance reaching >200 cm/ky (the impact of this observation on fault
interpretation is as discussed below).
In addition to the shelf edge belt, large thickness of Unit 4 is observed in a deep half-
graben separating a prominent dome at the center of the Dor disturbance from the shelf
edge (Fig. 4b). The accommodation space created by this half-graben is quickly filled
by sediments arriving from the nearby shelf edge. South of the Dor disturbance, the
half-graben is separated from the shelf edge (Fig. 5b). North of the disturbance, the two
features create a continuous sedimentary package (Fig. 5a). Noteworthy, the listric
faults east of the half-graben are different from all the other faults as will be discussed
below.

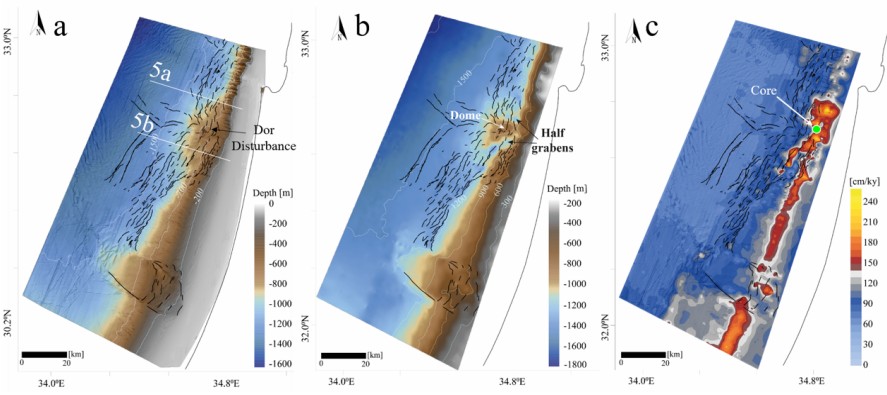

*Figure 4: (a) Faults on bathymetry background (After Tibor et al., 2013). (b) Base unit 4*
*structure map. (c) Unit 4 sedimentation rate. Half grabens separating the Dor disturbance from*
*the shelf edge and emphasizing its dome shape seen in b. These half grabens are filled with a*
*thick section of Unit 4 with sedimentation rate exceeding ~1.8 m/ky (c). High sedimentation*

*rate is also observed along the shelf edge expressing shelf progradation during the past 350*
*ky.*

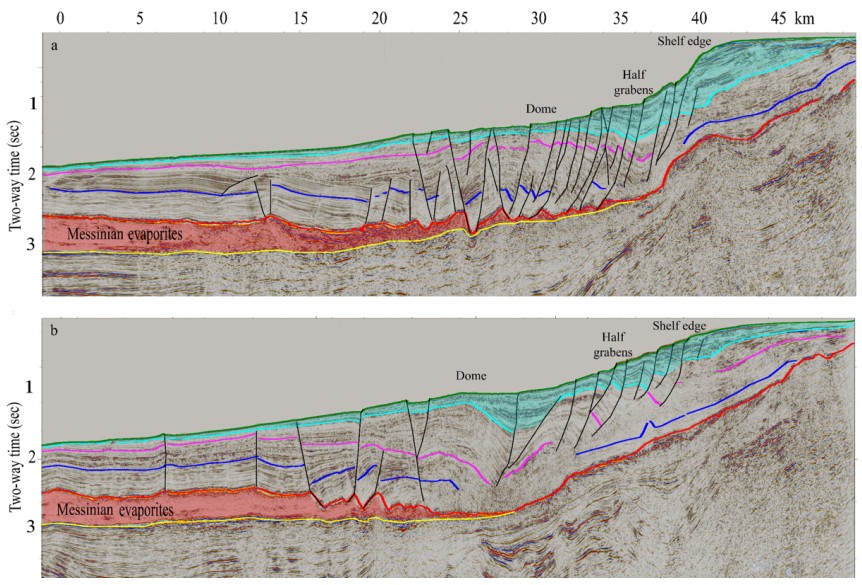

*Figure5 : (a) Cross-section north (a) and south (b) of the Dor disturbance (seismic location in Fig. 4a). Normal faults in black lines. Seismic reflectors as in Fig. 2.*



### *4.2. Fault vertical displacement*
To classify faults according to their vertical displacement, we assign each fault segment
a single value of maximum throw measured anywhere along its map trace (a) at the
seabed (height of scarp) and (b) at the base of Unit 4 (vertical offset). Results are
presented in Fig. 6 illustrating that seabed fault scarps higher than 30 m (red) are more
common near Dor and northwards (Fig. 6a), whereas fault scarps higher than 60 m
(turquoise) are observed only north of Dor (Fig. 6b) with the exception of one outlier
near the Palmahim Disturbance. This result is another illustration of the bathymetry
analysis presented above in Fig. 3a.
Consistent with our hypothesis that fault scarps are decreased by sedimentation and
erosion, classification according to vertical offsets at the base of Unit 4 (Fig. 6c,d)
portray a different picture with peak vertical displacements only in the vicinity of the
Dor disturbance (again, one outlier near Palmahim). In particular, we highlight the large
throws (>100 m) bounding the Dor disturbance from east (Fig. 6d), which partly
coincide with the listric faults mentioned above (Fig. 5). Uncommonly, these faults
form seabed scarps higher than 60 m (Fig. 6b) despite the exceptionally high
sedimentation rate observed at that location (Fig. 4c).

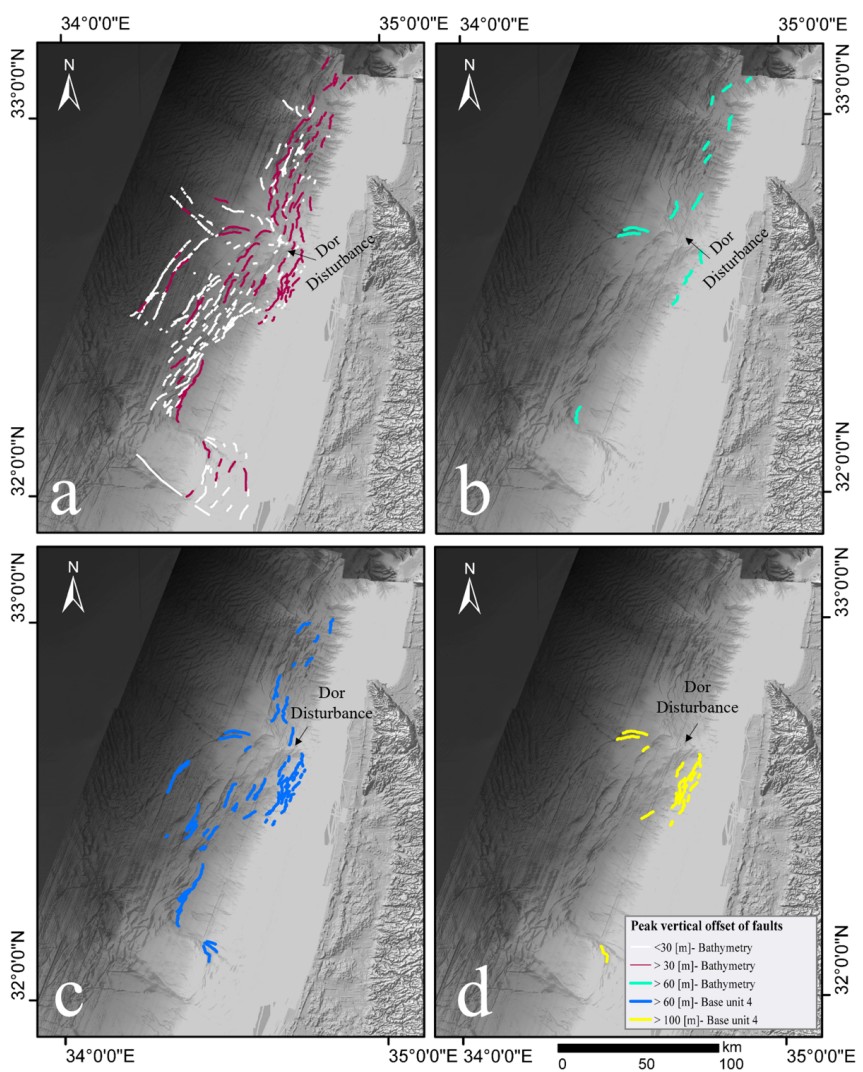


*Figure6 : Fault classification by vertical throw after assigning each fault segment with a single value representing the maximum vertical displacement measured anywhere along it. (a) Faults forming seabed scarps smaller (white) and higher (red) than 30 meters. (b) Faults forming seabed scarps larger than 60 meters (turquoise). (c) Faults displacing base Unit 4 by more than 60 m (blue). (d) Faults displacing base Unit 4 by more than 100 m (yellow). Note that faults with the largest vertical throw are concentrated around the Dor Disturbance. Background in all maps is shaded relief of bathymetry (Tibor et al., 2013).*


### 4.3. Fault planes and hidden faults

To map faults planes in the subsurface and measure their area, we use high-resolution 3D seismic volumes. Fig. 7 illustrates that 35 fault segments rupturing the seabed in the eastern side of the Sara-Myra survey, converge at depth to seven major faults. Noteworthy, a part of the fault marked by red (Fig. 7b) has no surface expression (Fig. 7a). This hidden fault ruptures the three lower horizons (Fig. 7d-f) reaching base Unit 4 in several locations (Fig. 7c) and unseen at the seabed (Fig. 7a). Similar analysis conducted for the Yam Hadera survey, illustrates that several major faults (marked green, purple, and yellow) are hidden (Fig. 8).

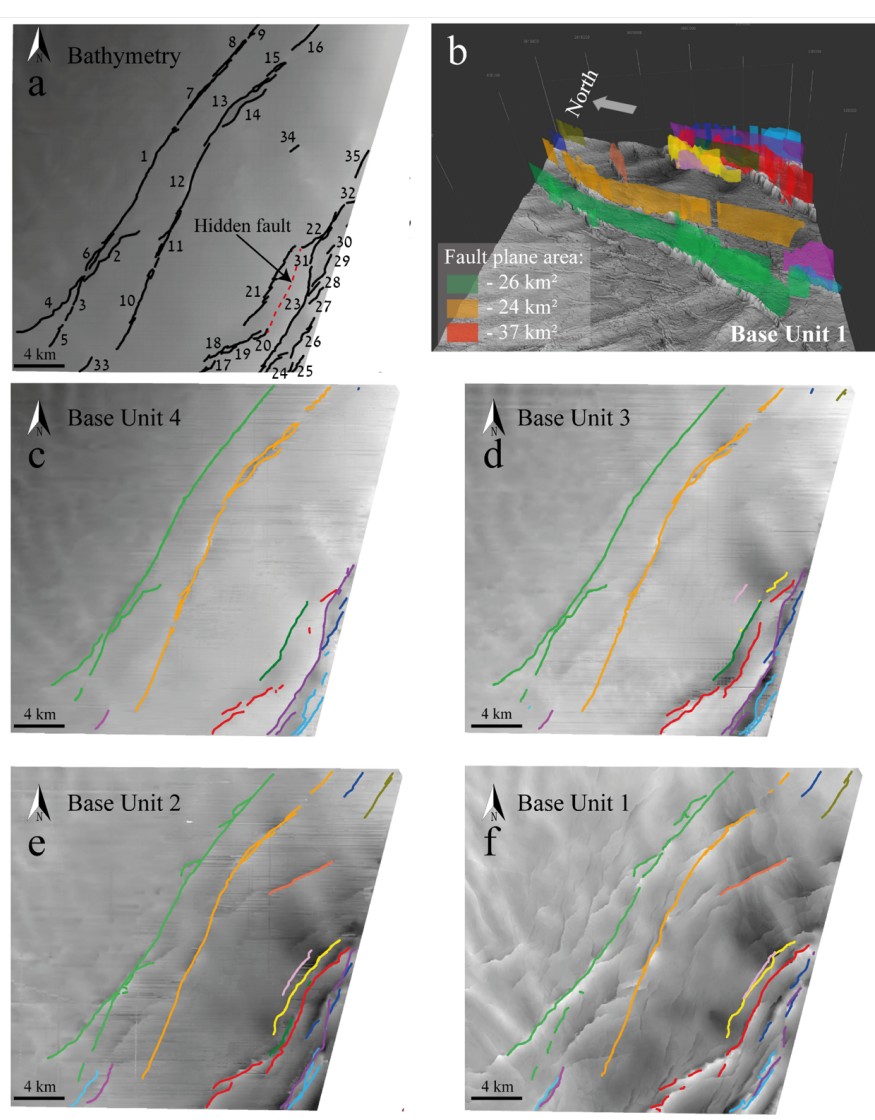

Figure 7: Subsurface mapping of fault planes. (a) 35 faults segments rupturing the seabed in
the eastern part of the Sara-Myra survey (location in Fig. 1). (b) A 3D view of fault planes
illustrating that the 35 fault segments at the seabed belong to 7 major major faults. plane areas
of those faults are measured. (c-f) Structural maps of four subsurface horizons (base units 4-
1), each with faults crossing it. Note the hidden faults (dashed black line in a), which do not
disrupt the seabed, but may rupture it in the future.

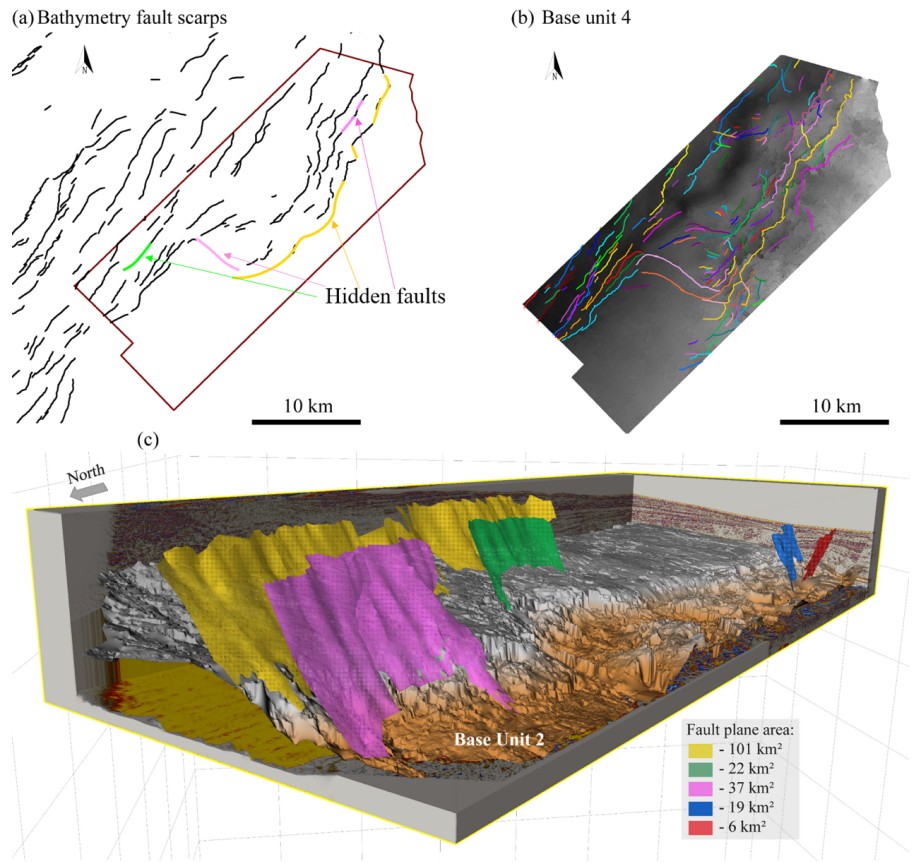

*Figure 8 : (a) Seabed faults in the Yam Hadera survey with hidden faults marked in color. (b)*
*All faults displacing base Unit 4. (c) 3D illustration of 5 faults with their measured fault plane*
*area. Note that the yellow and the pink faults are not detected at the seabed in some parts*
*(hidden faults) despite their large plane area ($101^2$, $37^2$ km, respectively).*







### 4.4. Fault geometry and location

A third way for fault classification is based on their geometry and location relative to
the underlying salt layer (Fig. 9). Group I produce horsts and grabens (marked blue)
mostly along the base of the continental slope, west of the salt wedge-out line. The
faults of Group I are located above a mobile salt layer. They displace the entire
Pliocene–Quaternary section down to the top salt horizon (Fig. 10, cross-section aa'),
and their dip angle varies around $45^0$ (Fig. 11).
Group II consists of seaward dipping faults producing a series of down-stepping stairs
(growth faults and half grabens) mainly in the upper slope, east of the salt wedge-out
line (Fig. 9). These faults are highly listric (Fig. 10, cross-section bb') as already
described above (Fig. 5). They are characterized by smaller dip angles of about $30^0$
(Fig. 11) and do not displace Unit 1 (Fig. 10 section bb').
Group III are relatively long strike-slip faults with a few hundred meters of lateral
displacement as demonstrated by Ben-Zeev and Gvirtzman, (2020). Their vertical
throw is much smaller and its direction changes along strike (Fig. 9).

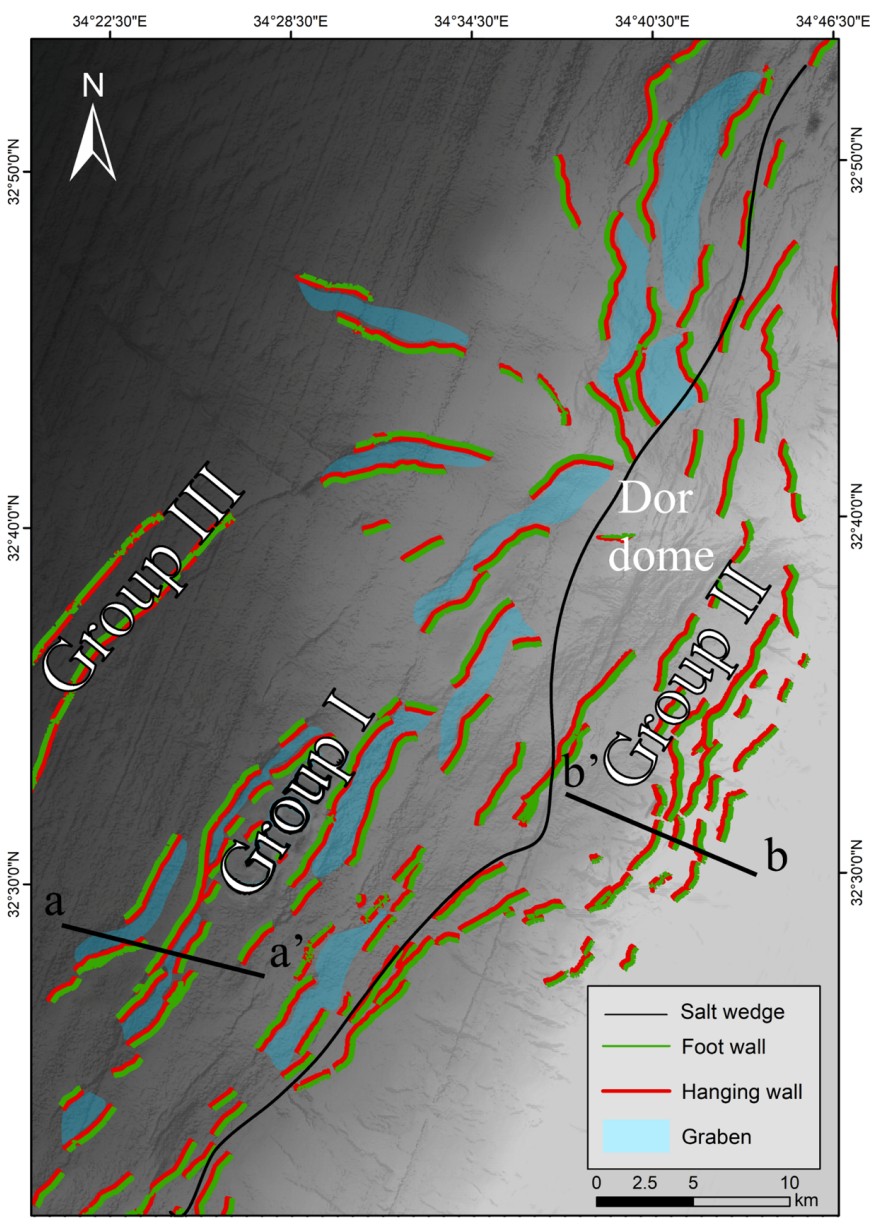


*Figure 9: Classification of faults according to structure and location. Hanging wall in red, foot*
*wall in green, and grabens in blue. Group I consists of horsts and grabens, running along the*
*base of the continental slope west of the salt wedge-out boundary (black lines). Group II*
*consists of down-stepping normal faults with hanging walls always in the basinward side,*
*mostly located east of the salt wedge-out line. Group III are strike-slip faults. Bathymetry from*
*Tibor et al. (2013).*





The high-resolution seismic volumes "Sara-Myra" and "Yam Hadera" allow detailed
investigation of the difference between Group I and Group II. Fig. 11 illustrates that the
vertical bathymetric offset (seabed scarp) is negatively correlated with the dip angle
(larger offsets for gently dipping faults), and positively correlated with the length of
fault planes measured in side view (larger offsets for longer faults). Moreover, Group
I, located in relatively deeper waters (yellow dots), is characterized by small (<15 m)
surface displacements (seabed scarps), high dip angles (>45$^0$), and relatively short faults
(0.5-2 km in a side view). Group II (the listric faults), located in shallower waters (blue
dots), is characterized by larger bathymetric displacements (15-35 m), lower dip angles
($\sim$30$^0$), and longer faults (1.5-4.5 km in side view). These observations highlight the
listric faults (Group II), located east of the salt pinch-out line, which are big in the
subsurface and also in their surface expression relative to Group I.

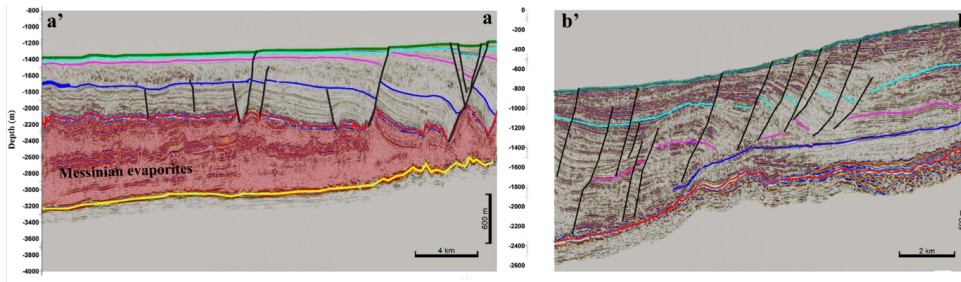


*Figure10 : Seismic cross-sections illustrating the difference between Group I (a-a', Sara-Mira*
*survey) located above the salt wedge and Group II (b-b', Yam-Hadera survey) located on the*
*continental slope east of the salt wedge. Note that faults of Group II do not displace Unit 1.*
*Location in Fig. 9.*


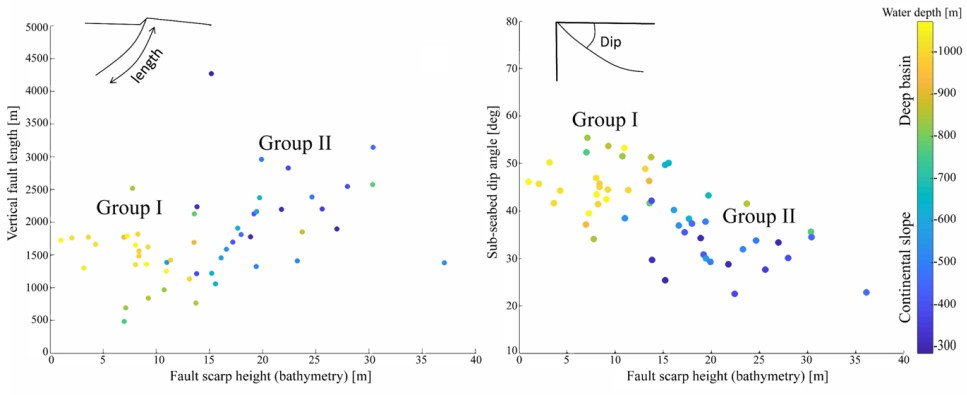

*Figure11 : Relationship between seabed scarp height, faults dip angle, vertical fault length*
*(i.e., length in cross section view), and water depth, in the "Yam Hadera" survey. Group II*
*located in the upper slope is characterized by larger vertical fault length (1.5-4.5 km) and*
*gently dipping (~30⁰) fault planes with larger fault scarps. As explained in the text, the large*
*throws in Group II are even more pronounced at the base of Unit 4.*


## 5. Discussion

### *5.1. Seabed versus subsurface mapping of faults*

Detailed mapping of the seafloor has become standard practice in marine geohazards

assessment and the demand for improved resolution is continuously growing. Here we

show that bathymetry is not enough for faults investigation even if it is extremely

detailed, because fault scarps are strongly affected by sedimentation and erosion. In

fact, subsurface mapping may be more informative even if its resolution is lower. For

example, peak vertical displacements of faults near the Dor disturbance are twice the

size of those measured along nearby faults; yet this is not observed on the bathymetry,

because the scarps are quickly buried. Sedimentation rates averaged on 350 ky indicate

>200 cm/ky near the Dor disturbance (Fig. 4c). Moreover, a 6-m-long core retrieved

nearby (location in Fig. 4c) with sedimentation rate of >850 cm/ky (Ashkenazi, 2021),

indicates that sedimentation rate may have increased recently.

The drawback of these measurements is their dependency on the quality of the seismic

data. Where only 2D lines are available, the measured value represents the throws at

the survey-fault intersection, which may represent the tip of the fault; moreover, some

faults may not be crossed by any seismic profile.

Additional support for the advantage of subsurface mapping is the structural map of the

350 ky horizon (Fig. 4b) and the sedimentation rate map (Fig. 4c). These maps show

that the most active regions in the study area are the half-grabens surrounding the Dor

disturbance from the east (Fig. 5). These half-grabens are rapidly subsiding (thick Unit

4) and the faults bounding them are the most active.

### *5.2. Fault classification*

Based on the maximal displacement of base Unit 4 (Fig. 6c,d), we classify all fault

segments mapped on the seabed (rupturing the seabed) to three vertical offset levels.



Vertical offset smaller than 60 m is considered low; 60-100 m is considered moderate;
and >100 m is considered high (Fig. 12a).
Based on the size (area of fault plane or its length on surface projection), we classify
all faults mapped at the subsurface to three levels. Fault planes smaller than 10 km$^2$ or
shorter than 5 km are considered small; area of 10-20 km$^2$ or length of 5-10 km is
considered moderate; and area larger than 20 km$^2$ or length longer than 10 km is
considered big (Fig. 12b).
It should be noted that unlike the classification by vertical displacement, which is
performed on seabed segments, the classification of faults by size is performed on fault
planes and a single fault plain frequently combines many seabed segments (i.e., the
number of fault planes in our database is significantly smaller than the number of seabed
segments).
Though the two classification criteria are independently measured, and despite a certain
degree of arbitrariness in choosing the cutoff values (60 m and 100 m of vertical
displacement; 10 km$^2$ and 20 km$^2$ for fault plane area), it is interesting to compare the
resulting maps. For most faults in the study area, the two criteria yield a similar category
(Fig. 12c,d). That is, faults segments with high displacement level are usually a part of
a big fault and vice versa. Exceptions, marked on Fig. 12 by black circles (moderate
displacement and small faults), mainly belong to Group II, which is exceptional in many
ways as shown above. Conversely, exceptions marked by red circles (big faults with
small displacement), belong to Group III, which are strike slip faults whose vertical
displacement is not expected to correlate with its dimensions.
Finally, we provide a simplified map that combines the two measured parameters to a
single hazard level (Fig. 13). In this map, high level is assigned to a fault segment,



which either is characterized by high displacement or belongs to a big fault; low means
low displacement and small size; moderate are all the rest.

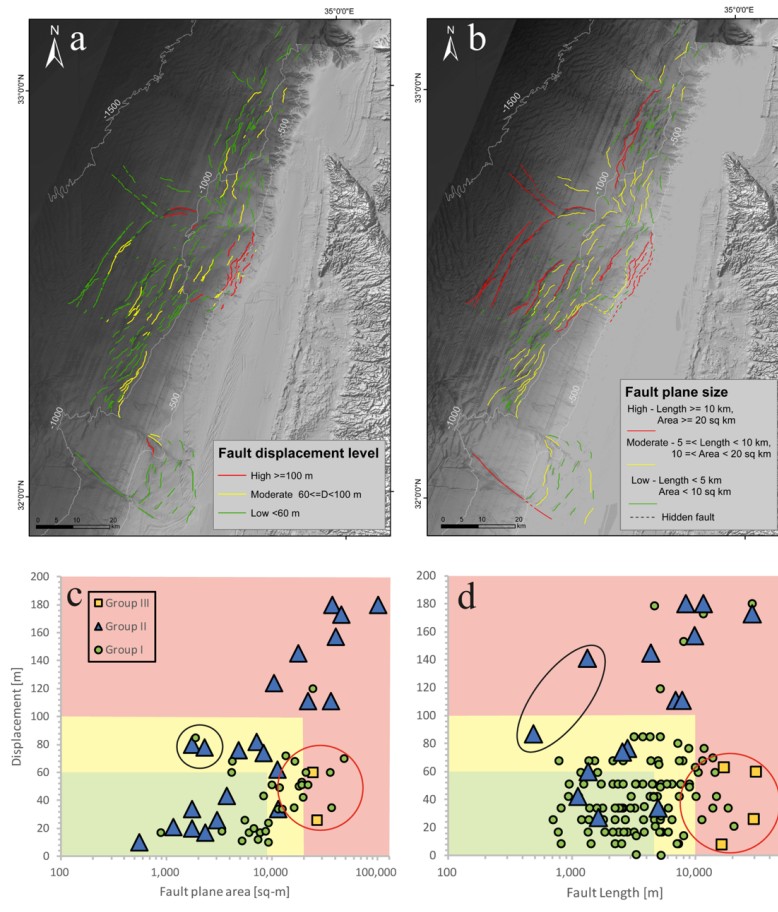


*Figure 12: Fault classification by displacement (a) and size (b). Each seabed fault segment is assigned a value based on its subsurface structure. i.e., the maximal displacement measured along the fault segment at the base unit 4 horizon and the total area of all segments, which connect at the subsurface. When 3D mapping of a fault is unavailable, fault size is expressed by its length in a map view. (c,d) Displacement at base Unit 4 versus fault size (length/area). Red, yellow, and green present three levels of displacement and size, which are proxies for surface rupture and potential earthquake magnitudes, respectively. While classification by the two criteria correlate for most faults, black circles mark faults that their displacement is high relative to their size, and red circles mark faults, which are big relative to their (vertical) displacement. Bathymetry from Tibor et al. (2013).*

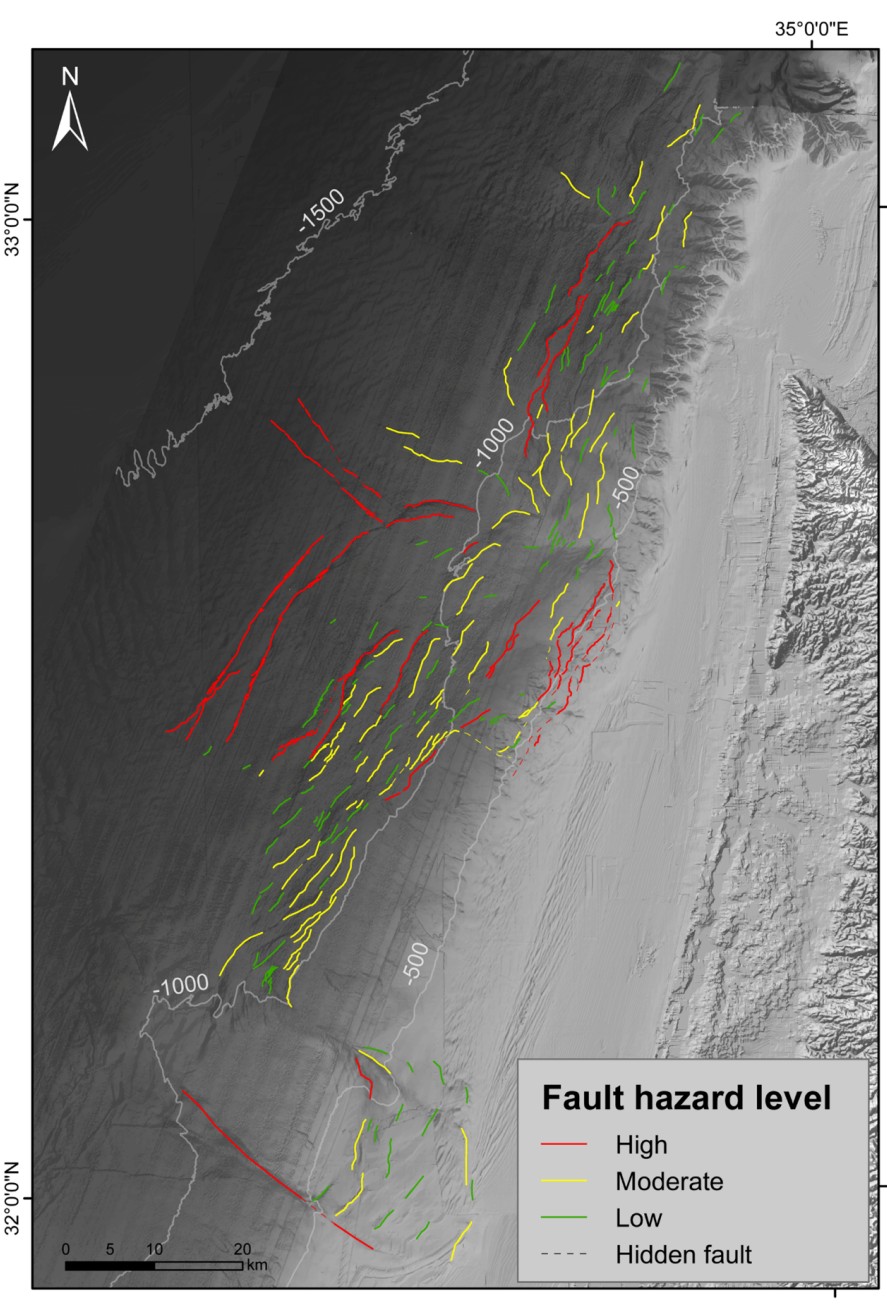

*Figure13 : Final simplified faults hazard map classified to three hazard level according to a combination of the two criteria presented in Fig. 12 (i.e., fault displacement and size). Combination is conservative., i.e., high level is assigned to a fault segment, which either is characterized by high displacement or belongs to a big fault; low means low displacement and small size; moderate are all the rest. Bathymetry from Tibor et al. (2013).*





### 5.3. *Listric faults south of the Dor disturbance*

The listric faults south of the Dor disturbance (part of Group II) are particularly
exceptional. Their planes dip gently with lower angles; they are long in a side view, but
do not penetrate Unit 1; they are located in the steep slope, east of the salt wedge; and
particularly important, they produce large seabed scarps despite their location in a high
sedimentation zone. In fact, sedimentation rate at that location is four times larger than
the displacement rate (~200 cm/ky vs. ~50 cm/ky, respectively. Fig. 3b, 4c). Allegedly,
this observation indicates that these faults are not continuously creeping, because if they
were creeping, sedimentation would continuously cover seabed scarps. Rather, we
argue, these faults jump seismically, and the scarps observed at the seabed today are
too young to be buried even by the rapid sedimentation. Such a possibility was raised
by Elfassi et al., (2019) for the deep basin faults of Group I in the Sara-Myra survey,
where sedimentation rates are similar or slightly higher than displacement rates. For the
listric faults described here, this conclusion is much stronger.

### 5.4. *Assessing the hazard of surface rupture*

The question which faults should be considered as active for hazard assessment has no
simple answer. The probability of fault to rupture the surface depends on many
parameters related to the seismic cycle: return period, cumulative stress since the last
seismic event, the ratio between slip and creep, stresses induced by nearby faults, and
more (Kiureghian and Ang, 1977).
Practically, the administrative definition of "active faults" for hazard mitigation on land
is largely based on data availability, that is, accurate mapping of faults traces on the
surface and poor knowledge of their subsurface continuation. In light of data
availability and social needs, many countries define active faults for hazard mitigation
as faults that have moved one or more times in the last 11,000 years (Bryant and Hart,
2007). Also, some countries use the category of "potentially active" for faults that



displace older markers (Kiureghian and Ang, 1977; Sagy et al., 2012) or geometrically
relate to active faults (Sagy et al., 2012).
Noteworthy, these definitions are binary - faults are either active or not – requiring no
probabilistic calculation. This approach for fault hazard mitigation is very different
from the approach for mitigating the damage from earthquake vibrations where the
probabilistic calculation of the ground motion is performed (e.g. Lermo and Chavez-
Garcia, 1993; Field and Jacob, 1995).
These two different approaches have led to different types of geological investigations.
For ground motion prediction, efforts are focused on determining magnitudes,
displacement rates, and return periods, whereas, for active faults, investigations are
focused on stratigraphic marker younger than ~11 ky to determine whether they are
displaced or not.
To determine whether each fault in the marine environment displaces a ~11 ky horizon,
we first need ultra-high-resolution seismic surveys aimed for a depth of tens of meters
to identify a suitable reflector; then, we need to drill, core, and date horizons in several
locations near each fault; doing that for large study areas may take a lifetime.
One practical option is to define all faults rupturing the seabed as active faults (On,
2016, USA). This approach is based on the rationale that if faults are identified at the
seabed despite sedimentation, they are likely active. However, note that fault scarps can
remain hundreds of thousands of years on the seabed without any additional jump when
the sedimentation rate is low.
In light of the difficulties of applying the onshore practice to the offshore environment,
we point out that the wealth of high-quality seismic data in the offshore area provides
opportunities that were never explored on land. Instead of focusing on high-resolution



bathymetry, we stress the importance of subsurface data. Our database (1) allows
identifying the amount of recent (in our case 350 ky) vertical displacement. (2) It allows
distinguishing between small seabed faults that are minor and small seabed faults that
are part of large faults. (3) It allows identification of hidden faults. (4) It allows
calculation of fault plane area.
The product of our analysis is a set of three maps. The first presents the recent vertical
displacement as a proxy for surface rupture (Fig. 12a); the second presents the size of
the fault as a proxy for potential magnitudes (Fig. 12b); the third generalizes the hazard
by combining the two proxies (Fig. 13). These maps do not aim to answer whether
faults are active or not, yet they are very useful for early planning of infrastructure
localities, because the highlight "red" and "green" zones.

## 6. Summary and conclusions

1. The need for geohazards assessment in the marine environment is increasing
globally. Yet, in the field of hazard maps for planning and building, the offshore
regions are commonly lagging decades behind the onshore practice.
2. Mapping 'active' faults in the marine environment is particularly complicated.
If the onshore practice is followed, a Holocene horizon is needs to be detected
in the subsurface; and then, for each fault the question, whether this horizon is
displaced or not needs to be answered. This requires high-resolution seismic
surveys and numerous coring and thus cannot be done for large regions.
3. In site-specific surveys, detailed bathymetry has become the main tool for
mapping faults. Yet, we demonstrate that this is not enough, because fault scarps
are decreased by sedimentation and erosion particularly in sediment rich
environments such as continental margins.



4. Here we take advantage of the marine environment (wealth of seismic data) to produce maps that cannot be produced onshore. First, we map a subsurface horizon dated to 350 ky in the entire study area. Second, we measure fault vertical displacements along this horizon. Third, we map fault planes combining several fault segments and measure their size.

5. By classifying all faults according to their vertical displacement and size, we prepare two hazard maps related to surface rupture and earthquake magnitudes, respectively. Then, we combine the two maps to one simplified fault map.

6. Our maps are particularly useful for master planning. The sedimentation rates map alone immediately reveals tectonically active grabens and the hazard maps help defining red and green zones.

7. Using our maps, we revealed a particularly problematic zone in the upper slope south of the Dor disturbance. In this area a series of big listric faults are characterized by large displacements. Sedimentation rate in this location is also exceptional - four times faster than displacement rate - and still, fault scarps are prominent. We suggest that this indicates seismic rupture rather than creep.

## 7. Author contribution

This study was conceptualized by ML under the supervision of ZG. Formal analysis, visualization of results and writing of the original draft were performed by ML. All authors contributed to the interpretation of the findings and revision of the paper.

## 8. Competing interests

The authors declare that they have no conflict of interest.



## 9. Acknowledgments

We are grateful to HIS Markit for providing us the Kingdom academic licenses for
seismic interpretation. Thanks to our colleagues in the subsurface laboratory at the
Geological Survey of Israel- Itzhak Hamdani, Yechiel Ben-Zeev, Jimmy Moneron,
Florencia Krawczyk, and Yael Sagy. This research was funded by the Israeli Ministry
of Energy, and the National Committee of Earthquake Preparedness and Mitigation.



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
