# Peer review of "Classifying offshore faults for hazard assessment: A new approach based on fault size and vertical displacement"

_Natural Hazards and Earth System Sciences, 2021_

## Referee Comment (RC2)

- 125

[referee-annotated manuscript omitted]

---

## Author Comment (AC1)

Nat. Hazards Earth Syst. Sci. Discuss., referee comment RC1
https://doi.org/10.5194/nhess-2021-393-RC1, 2022

[Figure]

**Comment on nhess-2021-393**

Anonymous Referee #1
* * *
Referee comment on "Classifying offshore faults for hazard assessment: A new approach based on fault size and vertical displacement" by May Laor and Zohar Gvirtzman, Nat. Hazards Earth Syst. Sci. Discuss., https://doi.org/10.5194/nhess-2021-393-RC1, 2022
* * *
Review to nhess-2021-393:

"Classifying offshore faults for hazard assessment: A new approach based on fault size and vertical displacement", by Laor and Gvirtzman.

Laor and Gvirtzman propose innovative and elegant approach to cope with fault hazards in marine environments, a challenging territory to explore this geohazard. The suggested methodology is formulated and exercised in a case study along the continental slope offshore Israel, but can be applied elsewhere around the world in similar marine environments. This paper is well worth publishing in NHESS.

Hereby I propose several comments and suggestions that in my opinion will improve the manuscript, widen the scope of the discussion, and extend its role among the previously published faults, landslides and seismicity maps of the study area.

We appreciate the positive evaluation of our paper and would like to thank Reviewer #1 for his helpful and comprehensive suggestions.

**General Comments**

- **Fault hazard**: The problematics of fault hazards should be explained already in the introduction so as to allow the reader a better background and understanding along the text and before arriving to section 5.4 and Figure 13. Please resolve this general term into its specific aspects: surface rupture, coseismic deformation and ground acceleration. Hidden/blind faults may produce coseismic deformation without surface break.

We think you are right. We will add a sentence in the introduction section, which explains what are the hazards posed by faults and emphasizes that this study focuses on surface rupture.

- **Fault maps**: Several fault maps have already been published in the past and besides citing them it is important to discuss, at least qualitatively, how the newly presented map relates to them. Furthermore, past researchers proposed hypotheses about specific faults, such as the one along the Israeli coastline, the Pelusium line (Neev et al,. 1973), transversal faults bordering the Palmahim disturbance (Garfunkel and Almagor, 1979), fault offshore the Carmel Coastal Plain (Kafri and Folkman, 1981), etc. I think it is important to place the present work and discuss its role among and along the history of research, at least in a quantitative manner.

Thank you for this important comment. Though this requires expansion of the geological background, we are willing to add a paragraph before section 2.2 that adds information from previous studies as follows:
In general, the Levant continental margin is consider tectonically-passive for more than 150 my since its formation in the early Mesozoic (Garfunkel, 1988). Nonetheless, Neev et al. (1973) raised the possibility that an active fault runs along the continental margin from offshore Lebanon to Sinai and named this fault "The Pelusium line". This suggestion, which may had a significant effect on seismic hazard estimations in Israel, produced a hot debate. Garfunkel at al. (1984) argued that all faults displacing the Plio-Quaternary section offshore Israel are related to salt tectonics and cannot produce significant earthquakes. Later on, aided by newer seismic material, Gvirtzman et al., (2008) and Gvirtzman and Steinberg (2012) showed that a series of deep seated faults were indeed active along the Pelusium line (continental margin fault zone) during the Oligocene and Early Miocene, when the Levant passive margin was reactivated alongside the Red Sea opening. Luckily, however, these faults stopped operating and are not active today. In contrast, north of the area studied here, deep-seated faults are operating alongside thin-skinned faults. The Carmel fault, located north of the area studied here, is an active brunch of the Dead Sea Transform (Karfi and Folkman, 1981); and active thrusting occurs offshore Lebanon (Elias et al., 2007).

- **Seismicity**: Studying active faults, there is a need to refer to the ongoing seismicity in the region (e.g. Katz and Hamiel, 2018) by discussing the finding of the present work in relation with the location, depth, magnitudes and mechanism of the continental slope seismicity, at least qualitatively.

Recently, Katz and Hamiel, (2018) showed that relocation of earthquakes offshore Israel indicates Mw<4 hypocenters at a depth of ~18 km along the continental margin fault zone mapped by Gvirtzman and Steinberg (2012). This finding is enigmatic, because these Miocene faults are covered by a few km of undisplaced rocks (Gvirtzman and Steinberg, 2012). One possibility to reconcile the two observations is that relocation offshore is uncertain and these small earthquakes may occur on shallow, salt-tectonics, faults rather than at depth of 18 km. Alternatively, maybe the Miocene faults that apparently stopped moving are still producing earthquakes. An active example may be the Suez Rift, which also (almost) stopped operating after the Miocene, but is still producing earthquakes.

- **Seismogenic zone**: The PGA map of the Israeli Building Code 413 is based on seismogenic zones defined by Shamir et al. (2001). How does the presented hazard map (e.g. Figure 13) relates to these zones? Should the continental slope be added as a new seismogenic zone to the database of the Israeli PGA map?

This question should be considered by scientists that will produce the next PGA map. Obviously, they will need to address the question of the potential magnitude. The question regarding the seismicity of the continental margin fault zone (or Pelusium line) is out of the scope of our study. The question regarding the potential magnitude of salt-related faults deserves more study and we intend to dig into it in the near future.

- **Landslides**: Same idea as above.

Thank you for this comment. We thought and debated this a lot and finally decided that we prefer not to get into the marine landslides subject because this paper focuses on faults hazard.  Nonetheless, we do cite Katz et al., (2015), in this context.

**Specific comments**

**Highlights**

I suggest rephrasing the highlights to better speak in favor of the importance, finding and potential application of this work. For example, the first highlight (Mapping "active faults"…) is a general notion not specific to this study; the forth highlight (Large faults scarps…) seems to have already been attributed to Elfassi et al. (2019a) in lines 142-144?

Thank you. After correcting the text, we will reconsider the phrasing of the highlights.

**Abstract**

You propose a new innovative approach and exemplify it on the specific case study of the Israeli continental slope. Why not wrapping up the abstract by proposing its implication and application to elsewhere similar marine environments, marine building codes, hazard assessment for submarine infrastructure facilities, etc?

We will add a sentence that clarifies the advantage of this map for early master planning and infrastructure route selection.

Line 21: Please explain in short, what do you mean by 'active faults': are they capable of surface rupture, coseismic surface deformation, ground acceleration, and within a given time frame? See also the relevant comment above.

Very important. Thank you we will explain that.

Line 28-29 (and 64-65): You write about three hazard levels but mention only two? What would be the role of the middle category?

We will also write about the intermediate hazard level. Noteworthy, the question regarding how to use our new map by regulators and planners is not in our yard. We define three hazard levels and regulatory agencies will decide how to use this maps. They may demand that infrastructure will not cross red faults; they may decide that red faults requires site specific surveys; or many other possibilities of usage.

**Introduction**

Lines 45-46: Some of the works mentioned in the introduction did dealt with active faults (e.g. Armijo et al., 2005); also, there is very interesting work of Elias et al. (2007) regarding active historical seismogenic fault offshore Lebanon, I think it should be mentioned as well.

Thank you. We will recheck the reference list.

The Dor and Palmahim disturbances play major role in this study. There is a need to give some background about them.

Our map relies on two criteria- (1) vertical offset, and (2) fault plane area/fault length in

map view, regardless of the tectonic mechanism of Dor and Palmahim disturbances. We prefer not to extant the geological background beyond what we wrote, and in particular that the second referee requested to shorten the background chapter.

Section 1.2 deals with the goal and the methodology of this work. Consider rephrasing the headline to 'Goal and methodology'?
Thank you, we will change it.

**Chapter 2. Scientific background**

Lines 144-147: I think this hypothesis needs to be verified by magnitude estimation. For example, as a thumb rule, M~6 crustal earthquakes are considered the minimum for generating surface rupture. What would be the estimated magnitude of the high (red) hazard class of faults for generating surface rupture - you have length, depth, area, and can assume vertical offset, say 1 meter?

This is a very good point and we thought about it a lot.

Observations:

1.  Surface rupture indicates M~6.

2.  Faults plane area indicates M<5 (figure 12)

3.  The Israeli earthquakes catalog consists of 2<M<4 (Katz and Hamiel, 2015).

    ⇨  Apparently, these thin-skinned salt-related faults do not follow the common ruls of thumb.
We are considering adding a paragraph on this topic in the discussion chapter, but not expanding on this complicated question that requires further deep research.

Lines 157-161: "… it has been suggested that faulting was initiated by basinwards salt flow" - is this explanation relevant also to group II (Figure 9) that is located outside the salt area? Or also to group I of strike slip nature?

Group II isn't related directly to the salt flow. However, we cannot reject the possibility of indirect relationships between Group I and Group II. Regarding group III (strike-slip nature)- we will add a reference describing the ss faults (Ben-Zeev and Gvirtzman, 2020).

Lines 171-174: There is a need to present in short the nature of the 350ky horizon, it is the key for evaluating the recent activity of the study faults. Similarly, describe in short the lithology of units 3 and 4. Is it the contrast between the two that yields the 350 ky horizon? Unit 4 is the lithological environment that hosts the faults system studied in this work.

The 350 ky horizon represents an unconformity, that is usually expressed in the form of a strong seismic reflector. Elfassi et al., 2019 described seismic units according to seismic facies. We do not have information about the lithology of the four seismic units except for the general notation that all units are part of Yafo formation, which consists mainly of clay and some sand. Our faults penetrate all 4 units.

**Section 3.2 Bathymetry data and Table 1**

What are the uncertainties associated with these grids, mainly in the vertical dimension, which is the key parameter to define the total offset and rate of slip.

We will add this information.

**Section 4.4. Fault geometry and location**

Lines 328-332: Looks to me also like a set of blocks rotated around horizontal axis?

Yes, but the location of the axis is not clear. We will add this term to the text.

**Discussion**

Line 380 – The very high sedimentation rate could also be attributed to down slope transport of materials?

Yes. Sedimentation includes all sources of material that are accumulated.
We will clarify this in the text.

**5.4. Assessing the hazard of surface rupture**

466-470: Please note that modern approach for surface rupture hazard mitigation is being developed towards Probabilistic Fault Displacement Hazard Analysis (PFDHA), much like PSHA for ground shaking.

Very important comment, we will mention it.

There are a few transversal (striking E-W) faults in the mapped region. They seem to be unique and deserve some attention.

We are not sure about the mechanism of these E-W transversal faults. The N-S transversal faults were mentioned in Ben-Zeev and Gvirtzman, (2020) and a mechanism was proposed. Also, we have a lack of 3D data on this area, compared to the N-S transversal faults area.

**Technical comments**

Hidden faults: Do you mean blind faults?

It may be the same, but these faults do not exactly meet the dry definition of "blind fault" and because of that we decided to call them "hidden faults".

In the blind fault ideal model, displacement decreases from a maximum located at the center of the fault plane to a tip line of zero displacements. Ideal blind faults grow by radial propagation with no migration of the point of maximum displacement, which is also the nucleation site of the fault (Watterson, 1986; Barnett et al., 1987). With the absence of dated horizons and knowledge about the phase of activity on these faults, it's difficult to differentiate between blind faults and syn-sedimentary faults. We can just say that they are not crossing all the youngest horizons but they do have a genetic relation to faults crossing the seabed.

Lines 243-249: Can you explain the reason for the increase of sedimentation rate from the deep basin towards the off shelf zone? If this area is also subject to slope failure, one would expect increase of sediment accumulation towards the basin?
This reflects the progradation of the shelf by all mechanisms including slope failure. In figure 4c you can see the progradation lineament with very high sedimentation rates along the shelf break.

Line 308-9: Please rephrase.

We rephrased it. Thank you.

Line 310: Should be: "dashed red line in…"?

Yes, definitely. Thank you.

**Figures**- We appreciate the comments on the figures and we will fix what is needed.

General: Please increase font size of coordinates, legend or any text where needed, e.g. Figures 3c, 4, 6, 9, etc.

Figure 1: I suggest increasing the area covered by the bottom left inset so as to allow orientation to readers who are not familiar with the research area, and please, add coastline.

Figure 3c: Please explain in the caption the areal extend of section 3c. Also, denote the location of Palmahim disturbance.

Figures 7, 8, 9: Please mark the location and extent of these maps on one of the previous figures.

Figure 10b: Please mark unit 1 on panel b.

**References and sources of Information**

Can you provide references or links to the sources of data mentioned in Table 1? Else, I believe you need to acknowledge these sources?

Reference to the Kingdom HIS platform?

**References not mentioned in the text**

Kafri, U. and Folkman, Y., 1981. Multiphase reverse vertical tectonic displacement across major faults in northern Israel. **Earth and Planetary Science Letters**. Volume 53, Issue 3, Pages 343-348

Katz, O. and Hamiel, Y., 2018. The nature of small to medium earthquakes along the Eastern Mediterranean passive continental margins, and their possible relationships to landslides and submarine salt-tectonic-related shallow faults. Geological Society, London, Special Publications, 477, 15-22, https://doi.org/10.1144/SP477.5

Shamir, G., Bartov, Y., Sneh, A., Fleischer, L., Arad, V. and Rosensaft, M., 2001, Preliminary seismic zonation in Israel. GSI Report No. GSI/12/2001, GII Report No. 550/95/01(1).

---

## Author Comment (AC2)

Nat. Hazards Earth Syst. Sci. Discuss., referee comment RC2
https://doi.org/10.5194/nhess-2021-393-RC2, 2022

[Figure]

**Comment on nhess-2021-393**

Stéphane Baize (Referee)

Referee comment on "Classifying offshore faults for hazard assessment: A new approach based on fault size and vertical displacement" by May Laor and Zohar Gvirtzman, Nat. Hazards Earth Syst. Sci. Discuss., https://doi.org/10.5194/nhess-2021-393-RC2, 2022

The paper presents a large compilation and analysis of high-quality bathymetric and seismic-reflection data, with the aim of providing classification and mapping of subsea faults that are potentially hazardous for installations.
The region of concern is vast (120 x 40 km²) and corresponds to part of the margin of the eastern Mediterranean basin, off Israel. This margin is subjected to significant deformations of the most superficial layers (0 to 1-2 km) of Plio-Quaternary age. These deformations are largely due to the mobility of this sedimentary cover on the salt layer dating from the Messinian, amplified by the slope of the margin.

The paper fits the objective of the journal, presents relevant products for further hazard analyses (surface displacement maps, deterministic hazard map) and then deserves to be published. However, I think there must be major revisions, including formal improvements, additional analysis, and moderation in some interpretation.

We appreciate the positive evaluation of our paper and would like to thank Reviewer #2 for his helpful and comprehensive suggestions for improving it.

The major criticisms are as follows.

- Some figures are too small and labels and captions are barely visible.
Thank you, we will enlarge the labels.

- The authors should develop the data and methods section and provide more information on data availability and processing options (mainly seismic reflection).
The data are industrial data that are part of the database of the Geological Survey of Israel. Unfortunately, Data availability is limited. Details can be obtained from the Israel Ministry of Energy (https://prime.energy.gov.il/).

- The part of Section 2 concerning ancient (pre-tertiary) geology could be lightened, and the part concerning elongated Neogene. In addition, a paragraph is missing which links (or not) the structures observed with the tectonic faults known on land.

We will add a paragraph describing previous works about on-land faults in section 2.2, as also requested by referee #1. We will add on-land faults on the inset map of figure 1.

- Section 4 (Results) should be restructured to be consistent with the maps and figures produced. Some figures are barely commented and deepened, which is a shame.

Thank you. We will improve and clarify the fig. captions and the text describing the results.

- Section 5 (Discussion) needs to be improved. Most of the current content is a summary of previous sections, not a discussion.

Combined with the requirements of Reviewer #1, we'll add a chapter at the end of the discussion that will discuss how the following observations may be reconciled:

1. Surface rupture

2. Fault plane area of thin-skinned faults

3. Separation between Miocene faults that have apparently ceased to be active and the thin-skinned faults.

4. Earthquakes location- in depth (Miocene), or in the shallow part (salt-related faults).

This discussion leads to the question of whether the relationship between surface rupture and magnitudes as recognized from crustal faults is valid for shallow, thin-skinned, faults rooted in salt.

- The interpretation of seabed scarps, as evidence of coseismic ruptures, is in my opinion doubtful, even erroneous, if one considers the context of the development of structures on the slope of the margin and the occurrence of salt-related deformations. To demonstrate a relationship with earthquakes, the authors must provide more observations.

Large (tens of meters) surface ruptures are observed in an area with high sedimentation (1-2 m/ky) rates. This necessarily means that the displacement rate is greater than the sedimentation rate, otherwise, all fault scarps would have been buried with no seabed expression. The fast motion of the faults may be explained by seismic ruptures, but the earthquake catalog does not support M~5 earthquakes predicted from the measured fault plane area of the big faults. Alternatively, the fast motion may express very fast creep (1-2 m/ky or more). This subject requires more research, which is out of the scope of this manuscript. However, in light of this comments, we will change the terminology from "seismic rupture" to "fast rupture", and add a discussion paragraph to clarify this enigmatic issue.

- There is a lack of analysis of the maps and displacement profiles produced (Figure 3 in particular), especially when comparing seabed scarp heights and Unit 4 throws at key locations, and/or for each fault identified.

All fault displacement values are projected on N-S profile. We will clarify this.

- It would be useful that the authors suggest ways of using the deterministic hazard map provided. Is this to be used to exclude any installation on «red» faults, or is it a decision tool for the engagement of further studies? Which of studies could be made to assess the hazard (probability of displacement)? What about the existing installations on "red", "yellow", "green" faults?

That is a major issue. We note, however, that the question of how to use this map in not in our field. We prepared this map for regulatory agencies and it is for them to define regulations. They may instruct infrastructure planners to choose a curvy route between faults; or to cross only "green" faults; or to require engineering solutions for sites crossing red faults. These decisions are not in the research field but in the regulatory field.

Specifically, in our case, the work was done with the support of the Ministry of Energy of Israel, and together with the Ministry of Environmental Protection and the Standards Institute will make decisions regarding the standards development.

- Finally, an important point is to state on the availability of original data (bathymetry, seismic profiles), developed tools (algorithm) and results (numerical files of fault maps, seabed scarps, displacement measurement points, etc.).
- The data are industrial data that are part of the database of the Geological Survey of Israel. Unfortunately, Data availability is limited. Details can be obtained from the Israel Ministry of Energy.

The details of the comments are available in the attached pdf file.

We appreciate the detailed comments that surely consumed time and effort. We will address them while correcting the manuscript. Thank you.

Stéphane Baize

Please also note the supplement to this comment:
https://nhess.copernicus.org/preprints/nhess-2021-393/nhess-2021-393-RC2-supplement.pdf

---

## Author Response (AR1)

Dear Editor and Reviewers,

We heartily thank the reviewers for their constructive and essential comments, which significantly improved the quality of our manuscript.

The main changes we made are:

1. We changed the paper's title so that it will be clear to the readers that the setting is offshore Israel.
2. We rewrote the **Abstract** according to the changes we made in the text to be more precise and informative.
3. We rewrote the **Introduction** section, so the problem of faults hazard is explained, common approaches are mentioned, and the need for this study is sharpened.
4. We added a new section (2.2) that presents former studies about the regional tectonic activity.
5. Following this new section, we added a sub-section to the **Discussion**, which discusses the meaning of deep-seated tectonic processes and their relations to the thin-skinned fault, which are the focus of this study.
6. We added a sub-section (2.4) introducing the Dor and Palmahim Disturbances.
7. We improved the **Data and Methods** section and added an intro paragraph, a sub-section describing the fault scarps mapping algorithm, and a sub-section describing the decisions and choices regarding mapping the subsurface faults.
8. In the **Results** section, we changed the titles and reordered the sub-sections according to reviewer #2 suggestions.
9. We improved the text, analysis, and figures according to the reviewer's comments.
10. We improved the **Discussion** section and added the 5.4 sub-section, as mentioned above.

These changes resolve the main problems highlighted by the two reviewers. In addition, we explained in detail all the specific questions and comments made by the reviewers in the two point-by-point response letters. These comments were implemented in numerous places along the text.

We are happy with all the changes we made as required by the two constructive reviewers, and we think that the new manuscript is now much better than the previous version.

Sincerely,

May Laor and Zohar Gvirtzman

[Figure]

Nat. Hazards Earth Syst. Sci. Discuss., referee comment RC1
https://doi.org/10.5194/nhess-2021-393-RC1, 2022

[Figure]

**Comment on nhess-2021-393**

Anonymous Referee #1
* * *
Referee comment on "Classifying offshore faults for hazard assessment: A new approach based on fault size and vertical displacement" by May Laor and Zohar Gvirtzman, Nat. Hazards Earth Syst. Sci. Discuss., https://doi.org/10.5194/nhess-2021-393-RC1, 2022
* * *
Review to nhess-2021-393:

"Classifying offshore faults for hazard assessment: A new approach based on fault sizeand vertical displacement", by Laor and Gvirtzman.

Laor and Gvirtzman propose innovative and elegant approach to cope with fault hazards inmarine environments, a challenging territory to explore this geohazard. The suggested methodology is formulated and exercised in a case study along the continental slope offshore Israel, but can be applied elsewhere around the world in similar marine environments. This paper is well worth publishing in NHESS.

Hereby I propose several comments and suggestions that in my opinion will improve themanuscript, widen the scope of the discussion, and extend its role among the previouslypublished faults, landslides and seismicity maps of the study area.

We appreciate the positive evaluation of our paper and would like to thank Reviewer #1 for his helpful and comprehensive suggestions.

**General Comments**

- **Fault hazard**: The problematics of fault hazards should be explained already in the introduction so as to allow the reader a better background and understanding along thetext and before arriving to section 5.4 and Figure 13. Please resolve this general term into its specific aspects: surface rupture, coseismic deformation and ground acceleration. Hidden/blind faults may produce coseismic deformation without surface break.

The whole **Introduction** was rewritten. Now the INTRO explains the problem of active faults and particularly the difficulty of the marine environment. We also added the possible approaches; (1) defining all seabed-crossing faults without dating as active, and (2) the probabilistic approach (PFDHA). We also explain the aim of this study to solve the situation in which a pipeline must cross a faulted zone to provide, in our case, gas onshore. We explain that here we deal only with surface rupture, but we added a discussion about the uncertainties related to earthquakes and ground motions.

- **Fault maps**: Several fault maps have already been published in the past and besides citing them it is important to discuss, at least qualitatively, how the newly presented map relates to them. Furthermore, past researchers proposed hypotheses about specific faults, such as the one along the Israeli coastline, the Pelusium line (Neev et al,. 1973), transversal faults bordering the Palmahim disturbance (Garfunkel and Almagor, 1979), fault offshore the Carmel Coastal Plain (Kafri and Folkman, 1981), etc.I think it is important to place the present work and discuss its role among and along the history of research, at least in a quantitative manner.

Thank you for this important comment. Though this requires expansion of the geological background, we added a paragraph numbered 2.2 that adds information from previous studies as follows:
In general, the Levant continental margin is considered tectonically passive for more than 150 Myr since its formation in the early Mesozoic (Garfunkel, 1988). Nonetheless, Neev et al. (1973) raised the possibility that an active fault runs along the continental margin from offshore Lebanon to Sinai and named this fault "The Pelusium line". This suggestion, which may have a significant effect on seismic hazard estimations in Israel, produced a hot debate. Garfunkel et al. (1984) argued that all faults displacing the Plio-Quaternary section offshore Israel are related to salt tectonics and cannot produce significant earthquakes. Later on, aided by newer seismic material, Gvirtzman et al. (2008) and Gvirtzman and Steinberg, (2012) showed that a series of deep-seated faults were indeed active along the Pelusium line (continental margin fault zone) during the Oligocene and Early Miocene when the Levant passive margin was reactivated alongside the Red Sea opening. Luckily, however, these faults stopped operating and are not active today. In contrast, north of the area studied here, deep-seated faults are operating alongside thin-skinned faults. The Carmel fault, located north of the area studied here, is an active branch of the Dead Sea Transform (Karfi and Folkman, 1981); and active thrusting occurs offshore Lebanon (Elias et al., 2007).

- **Seismicity**: Studying active faults, there is a need to refer to the ongoing seismicity in the region (e.g. Katz and Hamiel, 2018) by discussing the finding of the present work inrelation with the location, depth, magnitudes and

mechanism of the continental slope seismicity, at least qualitatively.

Recently, Katz and Hamiel, (2018) showed that relocation of earthquakes offshore Israel indicates Mw<4 hypocenters at a depth of ~18 km along the continental margin fault zone mapped by Gvirtzman and Steinberg (2012). This finding is enigmatic because these Miocene faults are covered by a few km of undisplaced rocks (Gvirtzman and Steinberg, 2012). One possibility to reconcile the two observations is that relocation offshore is uncertain, and these small earthquakes may occur on shallow, salt-tectonics faults rather than at a depth of 18 km.

In the corrected manuscript, this discussion is presented in a new section numbered 5.4, "earthquakes and faults".

- **Seismogenic zone**: The PGA map of the Israeli Building Code 413 is based on seismogenic zones defined by Shamir et al. (2001). How does the presented hazard map (e.g. Figure 13) relates to these zones? Should the continental slope be added asa new seismogenic zone to the database of the Israeli PGA map?

This question should be considered by scientists that will produce the next PGA map. Obviously, they will need to address the question of the potential magnitudes. The question regarding the seismicity of the continental margin fault zone (or Pelusium line) is out of the scope of our study. The question regarding the potential magnitude of salt-related faults deserves more study, and we intend to dig into it in the near future.

In the corrected MS we mention at the end that this issue needs more research.

- **Landslides**: Same idea as above.

Thank you for this comment. We thought and discussed this a lot and finally decided that we prefer not to get into the marine landslides subject because this paper focuses on faults hazard. Nonetheless, we do cite Katz et al., (2015), in this context.

**Specific comments**

**Highlights**

I suggest rephrasing the highlights to better speak in favor of the importance, finding andpotential application of this work. For example, the first highlight (Mapping "active faults"…) is a general notion not specific to this study; the forth highlight (Large faults scarps…) seems to have already been attributed to Elfassi et al. (2019a) in lines 142-144?

Thank you. We noticed that there are no highlights in this journal. Instead, we wrote a short summary.

**Abstract**

You propose a new innovative approach and exemplify it on the specific case study of theIsraeli continental slope. Why not wrapping up the abstract by proposing its implication and application to elsewhere similar marine environments, marine building codes, hazardassessment for submarine infrastructure facilities, etc?

We emphasize the aim of our study is to provide a map for early master planning and infrastructure route selection.

Line 21: Please explain in short, what do you mean by 'active faults': are they capable ofsurface rupture, coseismic surface deformation, ground acceleration, and within a given time frame? See also the relevant comment above.

This is explained in the new introduction.

Line 28-29 (and 64-65): You write about three hazard levels but mention only two? Whatwould be the role of the middle category?

We rephrased this in the corrected MS.

Noteworthy, the question regarding how to use our map by regulators and planners is not in our yard. We define three hazard levels, and regulatory agencies will decide how to use these maps. They may demand that infrastructure will not cross high-hazard level faults; they may decide that high-hazard level faults require site-specific surveys; or many other possibilities of usage.

**Introduction**

Lines 45-46: Some of the works mentioned in the introduction did dealt with active faults(e.g. Armijo et al., 2005); also, there is very interesting work of Elias et al. (2007) regarding active historical seismogenic fault offshore Lebanon, I think it should be mentioned as well.

Thank you. We rephrased and added references.

The Dor and Palmahim disturbances play major role in this study. There is a need to givesome background about them.

We added section #2.4 to the geological background about Palmahim and Dor disturbances.

Section 1.2 deals with the goal and the methodology of this work. Consider rephrasing theheadline to 'Goal and methodology'?

We included the goal in the INTRODUCTION section.

**Chapter 2. Scientific background**

Lines 144-147: I think this hypothesis needs to be verified by magnitude estimation. Forexample, as a thumb rule, M~6 crustal earthquakes are considered the minimum for generating surface rupture. What would be the estimated magnitude of the high (red) hazard class of faults for generating surface rupture - you have length, depth, area, andcan assume vertical offset, say 1 meter?

We now discuss earthquakes and faulting in the last paragraph of the discussion. We mention Katz and Hamiel, (2019), Wetzler and Kurzon, (2016), and Wells and Coppersmith, (1994).

Lines 157-161: "… it has been suggested that faulting was initiated by basinwards salt flow" - is this explanation relevant also to group II (Figure 9) that is located outside thesalt area? Or also to group I of strike slip nature?

Group II is not related directly to the salt flow. However, we cannot reject the possibility of indirect relationships between Group I and Group II. Regarding group III (strike-slip nature)- we added a reference describing the strike-slip faults (Ben-Zeev and Gvirtzman, 2020).

Lines 171-174: There is a need to present in short the nature of the 350ky horizon, it is the key for evaluating the recent activity of the study faults. Similarly, describe in short the lithology of units 3 and 4. Is it the contrast between the two that yields the 350 ky horizon? Unit 4 is the lithological environment that hosts the faults system studied in thiswork.

The 350 ky horizon represents an unconformity that is usually expressed in the form of a strong seismic reflector. Elfassi et al. (2019) described seismic units according to seismic facies. We do not have information about the lithology of the four seismic units except for the general notation that all units are part of the Yafo formation, which consists mainly of clay and some sand. Our faults penetrate all 4 units.

**Section 3.2 Bathymetry data and Table 1**

What are the uncertainties associated with these grids, mainly in the vertical dimension,which is the key parameter to define the total offset and rate of slip.

We added this information in table 1.

**Section 4.4. Fault geometry and location**

Lines 328-332: Looks to me also like a set of blocks rotated around horizontal axis?

We added the term rotated blocks in section 4.2.3. Thank you.

**Discussion**

Line 380 – The very high sedimentation rate could also be attributed to down slopetransport of materials?
Yes. Sedimentation includes all sources of material that are accumulated. We clarify this in the text.

**5.4. Assessing the hazard of surface rupture**

466-470: Please note that modern approach for surface rupture hazard mitigation is beingdeveloped towards Probabilistic Fault Displacement Hazard Analysis (PFDHA), much like PSHA for ground shaking.

Very important comment. Thanks. We added this in the introduction.

There are a few transversal (striking E-W) faults in the mapped region. They seem to beunique and deserve some attention.

We are not sure about the mechanism of these E-W transversal faults. The N-S transversal faults were mentioned in Ben-Zeev and Gvirtzman, (2020), and a mechanism was proposed. Also, we have a lack of 3D data on this area compared to the N-S transversal faults area.

**Technical comments**

Hidden faults: Do you mean blind faults?

It may be the same, but these faults do not exactly meet the dry definition of "blind fault" and because of that, we decided to call them "hidden fault segments".

In the blind fault ideal model, displacement decreases from a maximum located at the center of the fault plane to a tip line of zero displacements. Ideal blind faults grow by radial propagation with no migration of the point of maximum displacement, which is also the nucleation site of the fault (Watterson, 1986; Barnett et al., 1987). With the absence of dated horizons and knowledge about the phase of activity on these faults, it's difficult to differentiate between blind faults and syn-sedimentary faults. We can just say that they are not crossing all the youngest horizons, but they do have a genetic relation to faults crossing the seabed. Also, we changed the term from "hidden fault" to "hidden fault segments" to clarify we are relating only to fault segments that connect to segments that cross the seabed.

Lines 243-249: Can you explain the reason for the increase of sedimentation rate from thedeep basin towards the off shelf zone? If this area is also subject to slope failure, one would expect increase of sediment accumulation towards the basin?
This reflects the progradation of the shelf by all mechanisms, including slope failure. In figure 4c you can see the progradation lineament with very high sedimentation rates along the shelf break.

Line 308-9: Please rephrase.

We rephrased it. Thank you.

Line 310: Should be: "dashed red line in…"?

Yes, definitely. Thank you.

**Figures**- We appreciate the comments on the figures, and we fixed what is needed.

General: Please increase font size of coordinates, legend or any text where needed, e.g.Figures 3c, 4, 6, 9, etc.

Figure 1: I suggest increasing the area covered by the bottom left inset so as to alloworientation to readers who are not familiar with the research area, and please, add coastline.

Figure 3c: Please explain in the caption the areal extend of section 3c. Also, denote thelocation of Palmahim disturbance.

Figures 7, 8, 9: Please mark the location and extent of these maps on one of the previousfigures.

Figure 10b: Please mark unit 1 on panel b.

**References and sources of Information**

Can you provide references or links to the sources of data mentioned in Table 1? Else, Ibelieve you need to acknowledge these sources?

Reference to the Kingdom HIS platform?

We added.

**References not mentioned in the text**

Kafri, U. and Folkman, Y., 1981. Multiphase reverse vertical tectonic displacement across major faults in northern Israel. **Earth and Planetary Science Letters**. Volume 53, Issue3, Pages 343-348

Katz, O. and Hamiel, Y., 2018. The nature of small to medium earthquakes along the Eastern Mediterranean passive continental margins, and their possible relationships to landslides and submarine salt-tectonic-related shallow faults. Geological Society, London,Special Publications, 477, 15-22, https://doi.org/10.1144/SP477.5

Shamir, G., Bartov, Y., Sneh, A., Fleischer, L., Arad, V. and Rosensaft, M., 2001, Preliminary seismic zonation in Israel. GSI Report No. GSI/12/2001, GII Report No.550/95/01(1).

We added.

Thank you,

May Laor and Zohar Gvirtzman

[Figure]

Nat. Hazards Earth Syst. Sci. Discuss., referee comment RC2
https://doi.org/10.5194/nhess-2021-393-RC2, 2022

[Figure]

**Comment on nhess-2021-393**

Stéphane Baize (Referee)

Referee comment on "Classifying offshore faults for hazard assessment: A new approach based on fault size and vertical displacement" by May Laor and Zohar Gvirtzman, Nat. Hazards Earth Syst. Sci. Discuss., https://doi.org/10.5194/nhess-2021-393-RC2, 2022

The paper presents a large compilation and analysis of high-quality bathymetric and seismic-reflection data, with the aim of providing classification and mapping of subsea faults that are potentially hazardous for installations.
The region of concern is vast (120 x 40 km²) and corresponds to part of the margin of the eastern Mediterranean basin, off Israel. This margin is subjected to significant deformations of the most superficial layers (0 to 1-2 km) of Plio-Quaternary age. These deformations are largely due to the mobility of this sedimentary cover on the salt layer dating from the Messinian, amplified by the slope of the margin.

The paper fits the objective of the journal, presents relevant products for further hazard analyses (surface displacement maps, deterministic hazard map) and then deserves to be published. However, I think there must be major revisions, including formal improvements, additional analysis, and moderation in some interpretation.

We appreciate the positive evaluation of our paper and would like to thank Reviewer #2 for his helpful and comprehensive suggestions for improving it.

The major criticisms are as follows.

- Some figures are too small and labels and captions are barely visible.

-
Thank you, we enlarged the labels.

- The authors should develop the data and methods section and provide more information on data availability and processing options (mainly seismic reflection).

The data are industrial data that are part of the database of the Geological Survey of Israel. Unfortunately, data availability is limited. Details can be obtained from the Israel Ministry of Energy (https://prime.energy.gov.il/).

- The part of Section 2 concerning ancient (pre-tertiary) geology could be lightened, and the part concerning elongated Neogene. In addition, a paragraph is missing which links (or not) the structures observed with the tectonic faults known on land.

We added a paragraph describing previous works about on-land faults in section 2.2, as also requested by referee #1. We also add on-land faults on the inset map of figure 1.

- Section 4 (Results) should be restructured to be consistent with the maps and figures produced. Some figures are barely commented and deepened, which is a shame.
-
Thank you. We improved and clarified the fig. captions and the text describing the results exactly as suggested. Thanks.

- Section 5 (Discussion) needs to be improved. Most of the current content is a summary of previous sections, not a discussion.

Combined with the requirements of Reviewer #1, we added section 5.4 at the end of the discussion that discusses how the following observations may be reconciled:

1. Surface rupture

2. Fault plane area of thin-skinned faults

3. The separation between Miocene faults that have apparently ceased to be active and the thin-skinned faults.

4. Earthquakes depth location- at Miocene depths or in the shallow part (salt-related faults).

This discussion leads to the question of whether the relationship between surface rupture and magnitudes as recognized from crustal faults is valid for shallow, thin-skinned faults rooted in salt.

- The interpretation of seabed scarps, as evidence of coseismic ruptures, is in my opinion doubtful, even erroneous, if one considers the context of the development of structures on the slope of the margin and the occurrence of salt-related deformations. To demonstrate a relationship with earthquakes, the authors must provide more observations.

-
Large (tens of meters) surface ruptures are observed in areas with high sedimentation (1-2 m/ky) rates. This necessarily means that the displacement rate is greater than the sedimentation rate. Otherwise, all fault scarps would have been buried with no seabed expression. The fast motion of the faults may be explained by seismic ruptures, but the earthquake catalog does not support M~5 earthquakes predicted from the measured fault plane area of the big faults. Alternatively, the fast motion may express very fast creep (1-2 m/ky or more). This subject requires more research, which is out of the scope of this manuscript. However, in light of these comments, we changed the terminology from "seismic rupture" to "episodic tremor and slip" and added a discussion paragraph to clarify this enigmatic issue.

- There is a lack of analysis of the maps and displacement profiles produced (Figure 3 in particular), especially when comparing seabed scarp heights and Unit 4 throws at key locations, and/or for each fault identified.

-
We improved our analysis and produced a new figure (#11). This figure answers some of your comments and suggestions regarding figure 3.

- It would be useful that the authors suggest ways of using the deterministic hazard map provided. Is this to be used to exclude any installation on «red» faults, or is it a decision tool for the engagement of further studies? Which of studies could be made to assess the hazard (probability of displacement)? What about the existing installations on "red", "yellow", "green" faults?

That is a major issue. We note, however, that the question of how to use this map is not

in our field. We prepared this map for regulatory agencies, and it is for them to define regulations. They may instruct infrastructure planners to choose a curvy route between faults, cross only "low hazard level" faults, or require engineering solutions for ,sites crossing red faults. These decisions are not in the research field but in the regulatory field.

Specifically, in our case, the work was done with the support of the Ministry of Energy of Israel, and together with the Ministry of Environmental Protection and the Standards Institute will make decisions regarding the standards development.

- Finally, an important point is to state on the availability of original data (bathymetry, seismic profiles), developed tools (algorithm) and results (numerical files of fault maps, seabed scarps, displacement measurement points, etc.).

- The data are industrial data that are part of the database of the Geological Survey of Israel. Unfortunately, Data availability is limited. Details can be obtained from the Israel Ministry of Energy.

The details of the comments are available in the attached pdf file.

We appreciate the detailed comments that surely consumed time and effort. We addressed them as follows:

1. We change the title of the paper, including "Israel".

2. "I don't understand in which extent your approach difers from this one…"

   The common approach for site-specific surveys aims to apply the on-land practice to the offshore environment by multiple coring and dating Holocene horizons. We suggest a completely different approach. We are not interested in the question of whether Holocene horizons are displaced. Instead, we classify faults into three hazard levels and let the planners choose the least hazardous routes.

3. We elaborated the data and method section.

4. We changed the term "hidden fault" to "hidden fault segments", and added a section explaining how we map these faults.

5. We explain that the choice of threshold values is for convenience and can be changed.

6. We elaborated the captions of Figures 7-8 and explained them in the text.

7. Figure 11 was remade. We changed it significantly. We think that now it is much better. Thanks.

Thank you,

May Laor and Zohar Gvirtzman